# Registered report: Age-preserved semantic memory and the CRUNCH effect manifested as differential semantic control networks: An fMRI study

**Niobe Haitas**[1,2]*, **Jade Dubuc**[2], **Camille Massé-Leblanc**[2], **Vincent Chamberland**[3], **Mahnoush Amiri**[1], **Tristan Glatard**[4], **Maximiliano Wilson**[5], **Yves Joanette**[1,2], **Jason Steffener**[6]

**1** Laboratory of Communication and Aging, Institut Universitaire de Gériatrie de Montréal, Montreal, Quebec, Canada, **2** Faculty of Medicine, Université de Montréal, Montreal, Quebec, Canada, **3** Faculty of Arts and Sciences, Université de Montréal, Montreal, Quebec, Canada, **4** Department of Computer Science and Software Engineering, Concordia University, Montreal, Quebec, Canada, **5** Centre de Recherche CERVO – CIUSSS de la Capitale-Nationale et Département de Réadaptation, Université Laval, Quebec City, Quebec, Canada, **6** Interdisciplinary School of Health Sciences, University of Ottawa, Ottawa, Ontario, Canada

* niobe.haitas@gmail.com

## Abstract

Semantic memory representations are generally well maintained in aging, whereas semantic control is thought to be more affected. To explain this phenomenon, this study tested the predictions of the Compensation-Related Utilization of Neural Circuits Hypothesis (CRUNCH), focusing on task demands in aging as a possible framework. The CRUNCH effect would manifest itself in semantic tasks through a compensatory increase in neural activation in semantic control network regions but only up to a certain threshold of task demands. This study compares 39 younger (20–35 years old) with 39 older participants (60–75 years old) in a triad-based semantic judgment task performed in an fMRI scanner while manipulating task demand levels (low versus high) through semantic distance. In line with the CRUNCH predictions, differences in neurofunctional activation and behavioral performance (accuracy and response times) were expected in younger versus older participants in the low- versus high-demand conditions, which should be manifested in semantic control Regions of Interest (ROIs). Our older participants had intact behavioral performance, as proposed in the literature for semantic memory tasks (maintained accuracy and slower response times (RTs)). Age-invariant behavioral performance in the older group compared to the younger one is necessary to test the CRUNCH predictions. The older adults were also characterized by high cognitive reserve, as our neuropsychological tests showed. Our behavioral results confirmed that our task successfully manipulated task demands: error rates, RTs and perceived difficulty increased with increasing task demands in both age groups. We did not find an interaction between age group and task demand, or a statistically significant difference in activation between the low- and high-demand conditions for either RTs or accuracy. As for brain activation, we did not find the expected age group by task demand interaction, or a significant main effect of task demand. Overall, our results are

**Data Availability Statement:** We have shared stimuli, behavioral data, preprocessed functional data sets in the MNI space and statistical maps publicly in the OSF (https://osf.io/) with a digital object identifier (DOI: 10.17605/OSF.IO/F2XW9).

**Funding:** This study was funded by CIHR. The funders had no role in study design, data collection and analysis, decision to publish, or preparation of the manuscript.

**Competing interests:** The authors have declared that no competing interests exist.

compatible with some neural activation in the semantic network and the semantic control network, largely in frontotemporoparietal regions. ROI analyses demonstrated significant effects (but no interactions) of task demand in the left and right inferior frontal gyrus, the left posterior middle temporal gyrus, the posterior inferior temporal gyrus and the prefrontal gyrus. Overall, our test did not confirm the CRUNCH predictions.

## Introduction

Numerous neurophysiological declines affect various cognitive domains in the aging brain [1–3]. Nevertheless, language in general is well preserved in aging [4] and semantic memory may even improve across the lifespan [5–8]. Compared with attention or memory, the relative preservation of language throughout the lifetime [9] may be justified by the need to maintain successful communication, resulting in compensatory, flexible or atypical recruitment of neural resources [1]. Accuracy in semantic tasks is generally well maintained in older adults given their more extensive experience with word use and larger vocabulary than younger adults [1, 5, 8, 10–13]. Response times (RTs), however, are often longer than in younger adults [10], possibly because older adults are slower at accessing and retrieving conceptual representations from their semantic store [14–16], engaging the required executive function resources [17], and making the necessary motor responses [18]. Aside from behavioral performance, findings reported in the literature about the neural correlates sustaining older adults' versus younger adults' semantic memory are often conflicting, depending on the task used, interindividual variability and the specific age group. Although several hypotheses regarding age-related neurofunctional reorganization (e.g., Hemispheric Asymmetry Reduction in Older Adults—HAROLD [19] and Posterior-Anterior Shift in Aging—PASA [20]) aim to explain how aging affects cognitive skills in general, it is still not clear how aging impacts the underlying pattern of activation sustaining semantic memory, given its relative preservation throughout the life course. Older adults' relatively preserved semantic memory when compared with other cognitive fields [9, 21, 22] may be partly explained by the proposed dual nature of the semantic memory system, as expressed within the controlled semantic cognition framework [23–26]. This study focuses on the preservation of semantic memory, defined as the "cognitive act of accessing stored knowledge about the world" [27] in aging, using a semantic judgment task that manipulates semantic control with two demand levels (low and high).

To account for conflicting findings in terms of brain activation during semantic memory tasks and the relative preservation of semantic memory in normal aging, a possible explanation is to consider it the result of adaptive mechanisms captured within the CRUNCH model (Compensation-Related Utilization of Neural Circuits Hypothesis) [28]. According to this hypothesis, the level of task demands impacts performance and neurofunctional activation in both younger and older individuals; the effects of aging can then be thought of as the expression of increasing task demands earlier than in younger adults. Accordingly, additional neural resources are recruited to attempt to compensate when faced with higher task requirements, echoing an aspect of how the aging process is manifested [29, 30]. Compensation is defined as "the cognition-enhancing recruitment of neural resources in response to relatively high cognitive demand" [29]. Age-related reorganization phenomena refer to reduced neural efficiency, also known as dedifferentiation, which results in decreased performance in older adults [31–34].

At the same time, and as part of the age-related neurofunctional reorganization, neural resources may migrate from the default mode network (DMN) toward more urgent task

requirements, which can be expressed as underactivation in areas subserving "redundant" tasks [28]. Indeed, the more task demands increase, the more DMN activation is expected to decrease; however, this ability to "silence" the DMN is reduced in older adults [35]. Both over- and underactivation are relevant terms referring to comparisons with the optimal patterns of activation seen in younger adults [28]. Although the CRUNCH model describes compensatory neural mechanisms, it is not without its limits. For older adults, the benefit of overactivation is thought to reach a threshold beyond which additional neural resources do not suffice; after that point, activation declines and performance deteriorates [28]. The relationship between task demands and fMRI activation has been described as an inverted U-shaped curve, and the curve of older adults is situated to the left of the curve of younger ones. In other words, older adults recruit additional neural resources at lower levels of task demands, reach a maximum limit and decrease in activity as task demands continue to increase, and this happens earlier than in younger ones (see Fig 3a in [29, 30]).

The CRUNCH hypothesis was conceived on the basis of evidence from a working memory study. Activation increased in the dorsolateral prefrontal cortex when accuracy was maintained and decreased when accuracy was compromised, depending on task demands or the number of items successfully retained [36, 37]. Another working memory study obtained congruent results, suggesting that older adults may achieve the same outcomes using different neural circuits or strategies to achieve age-matched performance [38]. However, the CRUNCH predictions have not been confirmed in more recent working memory studies. In a working memory study with three load conditions using functional near-infrared spectroscopy (fNIRS), activation in the younger participants progressively increased in the prefrontal cortex (PFC) as demands increased, while performance was maintained [39]. However, in the older adults, when performance was compromised during the most difficult condition, activation remained high in the PFC bilaterally. Similarly, in a visuospatial working memory task with four levels of task demands, the CRUNCH predictions were not supported [40]. Instead, an increase in activation was found in a large network (premotor, prefrontal, subcortical and visual regions); however, no "crunch" point after which activation decreased was found for the older group. Although older adults showed more activation across regions than younger ones at the higher task demands, at the group level this difference was not significant, challenging the CRUNCH prediction that demands and fMRI activation would interact.

Compatible with the CRUNCH expectations, increased activation in frontoparietal regions with relatively maintained performance has been reported in several language studies; however, the results are not always consistent. More specifically, in a discourse comprehension study using fNIRS, increased activation was found in the left dorsolateral prefrontal cortex in older adults, whereas their performance was mostly equal to that of their younger counterparts [41]. In a sentence comprehension study, increased activation was observed in both younger and older adults during the most complex sentences in regions such as the bilateral ventral inferior frontal gyrus (IFG)/anterior insula, bilateral middle frontal gyrus (MFG), bilateral middle temporal gyrus (MTG), and left inferior parietal lobe [42]. Older adults showed more activity than younger ones in the IFG bilaterally and the anterior insula in the difficult condition; however, accuracy was not maintained. Partially compatible with the CRUNCH, overactivation with maintained performance has also been observed in a picture naming study that manipulated task demands/inhibition [43]. When naming difficulty increased, both younger and older adults showed increased activation in bilateral regions such as the IFG, anterior cingulate gyri, pre-, post-central, supramarginal and angular gyri, together with maintained performance, while older adults' RTs did not significantly increase [43]. Fewer studies exist on how semantic memory is affected by increasing task demands, which is the focus of our study.

Given the large volume of concepts and processes involved, semantic memory relies on a widely distributed and interconnected, mainly left-lateralized core semantic network [17, 27, 44–46]; this network includes the anterior temporal lobes (ATL) bilaterally, which have been proposed to act as semantic hubs [47, 48]. It has been suggested that semantic memory is organized as a dual system composed of two distinctive but interacting systems, one specific to representations and one specific to cognitive-semantic control [25, 46, 49–53]. In other words, it is thought to include processes related to stored concept representations with their modality-specific features, which interact with control processes in charge of selecting, retrieving, manipulating and monitoring them for relevance and the specific context, while suppressing irrelevant information [24–26, 54–57]. Within the controlled semantic cognition framework [26], the semantic control network is significantly recruited during more complex tasks subserved by left-hemisphere regions such as the PFC, IFG, posterior middle temporal gyrus (pMTG), dorsal angular gyrus (dAG), dorsal anterior cingulate (dACC), and dorsal inferior parietal cortex (dIPC) [25, 26, 45, 46, 51, 53, 58, 59], potentially extending toward the right IFG and PFC when demands intensify further [46]. A very up-to-date and extensive meta-analysis of 925 peaks over 126 contrasts from 87 studies on semantic control and 257 on semantic memory found further evidence of the regions involved in semantic control, concluding that they were the left-lateralized IFG, pMTG, posterior inferior temporal gyrus (pITG), and dorsomedial prefrontal cortex (dmPFC) [24]. Regions related to semantic control are thought to largely overlap the neural correlates of the semantic network [24] but also with regions related to the "multiple-demand" frontoparietal cognitive control network involved in planning and regulating cognitive processes [26, 60].

Differential recruitment has been found for easy and harder semantic tasks in younger adults, including recruitment of semantic control regions for hard tasks. For example, in a study using transcranial magnetic stimulation (TMS) to examine the roles of the angular gyrus (AG) and the pMTG, participants were required to perform identity or thematic matchings that were either strongly or weakly associated, based on previously collected ratings; RTs were used as an indicator of association strength. Stimulation to the AG and the pMTG confirmed their roles in more automatic and more controlled retrieval, respectively [58]. An fMRI study used a triad-based semantic similarity judgment task to compare responses to concrete and abstract nouns (imageability) while also manipulating difficulty. Difficulty was rated with semantic similarity scores based on word ratings; a semantic similarity score was computed for every triad to classify it as easy or hard. Increased activations were found during the hard triads, and regardless of word imageability, in regions modulating attention and response monitoring, such as the cingulate sulcus bilaterally, medial superior frontal gyrus and left dorsal IFG [61]. In a triad-based synonym judgment task comparing concrete and abstract words, where triads were categorized as easy or difficult based on the respective RT in relation to the group mean, a main effect of difficulty was confirmed: increased activation was reported in the left temporal pole, left IFG and left MTG [62]. In a triad-based task where participants were requested to match words for color and semantic relation to probe more automatic or controlled semantic processing, respectively, greater activation was found in the IFG and intraparietal sulcus (IPS) during the more difficult triads based on color-matching. In general, accuracy was maintained across conditions but participants made more errors and had longer RTs in the "difficult" color condition, lending support to the concept of controlled semantic cognition [50]. There is therefore evidence that activation increases in semantic control regions when semantic processing demands increase, which could be attributed to the "matching" of task requirements with available neural resources, in line with CRUNCH predictions. In aging, although the system related to representations is thought to be well maintained, the system related to cognitive-semantic control is believed to be more affected [23]. This study

focuses on how the relation between semantic control network activation and increasing task demand is affected by aging.

The neural correlates sustaining semantic memory are thought to be largely age-invariant, with only small differences in neural recruitment as a function of age [16, 22, 63–66]. In a recent meta-analysis of 47 neuroimaging studies comparing younger and older people, more activation in semantic control regions was reported in older adults than in younger ones, while accuracy was found to be equal between the two groups [22]. Although this increase in activation could be attributed to compensation, it could also reflect age-related loss of neuronal specificity or efficiency [22]. Several studies report activation and performance results in line with the compensatory overactivation hypothesis. In a semantic judgment task, participants had to decide whether two words share a common feature (shape or color); their performance was categorized as better or worse based on a split from behavioral data [56]. In better-performing older adults, activation was increased relative to younger adults in control regions such as the inferior parietal and bilateral premotor cortex—regions that are important for executive functions and object visual processing; relative to poorer-performing older adults, activation increased in the premotor, inferior parietal and lateral occipital cortex in better-performing older adults. A further analysis for gray matter found that increased gray matter in the right precentral gyrus was associated with maintained performance [56]. In a semantic categorization study, older participants performed as accurately as the younger ones but had longer RTs. Their maintained performance was correlated with activation in a larger network than in younger adults, including parts of the semantic control network (such as the left frontal and superior parietal cortex, left anterior cingulate, right angular gyrus and right superior temporal cortex); this network was reportedly atypical and excluded the PFC [44].

Specific recruitment of the left IFG, which is believed to be in charge of top-down semantic control [45, 49, 51, 67], has been reported in association with the "difficult" condition in several studies. In a triad-based semantic judgment task evaluating words for rhyme, semantic and perceptual similarity, interaction and conjunction analyses revealed a significant interaction between age and the high-load semantic condition. Older adults overrecruited the control-related regions of the left IFG, left fusiform gyrus and posterior cingulate bilaterally, when competition demands increased; their accuracy was even better than that of their younger counterparts [66]. In a picture-naming task, older adults recruited larger frontal areas than younger ones in both hemispheres. Although recruitment of the IFG bilaterally—and not solely in the left hemisphere—was beneficial to older participants' performance, the recruitment of other right-hemisphere regions was negatively correlated with accuracy [16]. These authors provided support for the claim that the neural substrates of semantic memory representations are intact in older adults, whereas it is the executive aspects of language functions, including accessing and manipulating verbal information, that are most affected by aging [16]. In another study with younger adults only, which aimed to dissociate the role of the IFG in phonologically versus semantically cued word retrieval, the recruitment of anterior-dorsal parts of the LIFG was associated with high task demands in the semantic fluency condition, while performance was maintained [68].

Evidence therefore exists of a correlation between increased activation in semantic control regions and increased task demands, which could support the compensation account of semantic memory performance in both younger and older adults, and potentially reflect the ascending part of the U-shaped relation between fMRI activation and task demands. However, attributing a causal relation between increased activation in the semantic control network and compensation is not straightforward. Distinguishing between the compensation and dedifferentiation accounts can be challenging, as merely correlating brain activation with behavioral outcomes that claim compensation is occurring is methodologically incomplete [69, 70]. Many

studies do not manipulate or cannot compare task demands; thus, interpreting results that correlate neural activation with behavior can be confusing [53]. For example, in a study where task demands are lower, reorganization may be interpreted as compensatory when performance is maintained; on the other hand, when performance is more affected, it may be attributed to dedifferentiation. Numerous methodological caveats exist when one attempts to assign meaning a posteriori to age-related reorganization, given the observational nature of neuroscience but also the need for more robust methodological designs, including longitudinal studies that measure in-person changes, between-region comparisons and better analytical approaches (for a review, see [70]). Correlating increased activation with improved performance at a single point in time and attributing it to compensation would require additional measures, especially since compensation may be attempted or only partly successful [30, 71].

According to the CRUNCH theory, the compensatory increase in activation of semantic control regions should reach a plateau beyond which additional resources no longer benefit performance [28]. As such, reduced activation in cognitive control regions when semantic processing demands increase has also been reported. According to CRUNCH, this reduced activation could be interpreted as indicating that neural resources have reached their maximum capacity and are no longer sufficient to successfully sustain compensation for the task [28]. Indeed, the above-mentioned meta-analysis of 47 neuroimaging studies comparing activation in younger and older adults (mean age of younger participants: 26 years (SD = 4.1); mean age of older participants: 69.1 (SD = 4.7)) during semantic processing tasks also reported decreased activation in the older adults in typical semantic control regions in the left hemisphere (IFG, pMTG, ventral occipitotemporal regions and dIPC) together with increased activation in "multiple-demand network" regions in the right hemisphere (IFG, right superior frontal and parietal cortex including the middle frontal gyrus, dIPC and dACC) especially when performance was sub-optimal [22]. In a semantic judgment task of words (living versus non-living) with two levels of difficulty and four across-the-lifespan age groups, activation outside the core semantic network increased linearly with age and contralaterally toward the right hemisphere (right parietal cortex and middle frontal gyrus) in the easy condition, while accuracy was maintained [64]. In the difficult condition, however, RTs were slower and reduced activation was observed in older participants' semantic control regions, namely the frontal, parietal and cingulate cortex regions, suggesting an age-related decline in the brain's ability to respond to increasing task demands by mobilizing semantic control network resources [64].

Similarly, increased activation in right-lateralized semantic control regions was detrimental to performance in both younger and older participants in a word generation study manipulating for task difficulty [72]. Indeed, activation in the ventral IFG bilaterally was correlated with difficult items, as opposed to easier ones, and reduced performance irrespective of age. In a verbal fluency study by the same group using correlation analysis, a strong negative correlation was found between performance and activation in right inferior and middle frontal gyrus ROIs [73]. Older adults demonstrated more bilateral activation than younger ones, especially in the right inferior and middle frontal regions, whereas their performance during the semantic task was negatively impacted. However, this kind of increase in activation in the right-lateralized semantic control network, together with a drop in performance has not been consistently documented. For example, in a semantic judgment task on word concreteness using magnetic encephalography (MEG), older participants overactivated the right posterior middle temporal gyrus, inferior parietal lobule, angular gyrus and left ATL and underactivated the control-related left IPC as a result of increased task demands; their performance was equivalent to the younger adults', thus supporting the compensatory hypothesis [65]. According to CRUNCH, these findings could be interpreted as falling within the descending part of the inverted U-shaped relation between semantic processing demands and fMRI activation [29]:

after a certain difficulty threshold, available neural resources from the semantic or multiple-demand control network have reached their maximum capacity, leading to reduced activation and a decline in performance [30].

In summary, it seems that, depending on the semantic task used and its perceived or actual difficulty, both increased and decreased activations have been reported in the semantic control network, along with variations in performance. The relationship between neural activation, task difficulty and behavioral performance is not a straightforward one. It is possible that the neural correlates of semantic memory remain relatively invariant throughout aging when the task is perceived as easy. On the other hand, when task difficulty, or perceived difficulty, increases, activation and behavioral performance may either increase or decrease depending on the nature of the task and its perceived or actual difficulty, in line with CRUNCH. Accordingly, maintained performance could depend on the additional recruitment of semantic control network resources but only between certain difficulty thresholds, before which increasing activation is unnecessary or beneficial and after which performance declines.

## Alternatives to CRUNCH in accounting for age-related reorganization

Several alternative neurofunctional reorganization hypotheses have been reported to account for the change in general cognitive skills in aging (for reviews, see ([30, 74, 75]). Such accounts often refer to the engagement of compensatory mechanisms and redistribution of resources through overactivation or deactivation, often including in the PFC [28, 30]. For example, the HAROLD neurofunctional reorganization hypothesis refers to reduced hemispheric asymmetry in older adults with the objective of maintaining high performance [19]. To reduce asymmetry, brain activation may increase and/or decrease in certain brain areas by recruiting additional and alternative neuronal circuits from the contralateral hemisphere. The resulting reduction in asymmetry optimizes performance, whereas elderly adults who maintain a unilateral or asymmetrical activation pattern similar to that of younger people do not perform as well [19]. Several studies have recently challenged the accuracy of the HAROLD model [76, 77]. Neurofunctional reorganization has also been reported to occur intrahemispherically. The PASA (Posterior-Anterior Shift in Aging) phenomenon provides an example of this type of reorganization [78], describing an age-related shift in activation from the occipitotemporal to the frontal cortex [20, 79]. PASA is considered to reflect a general age-related compensation phenomenon whereby sensory deficits are managed by decreasing activation in occipitotemporal regions and increasing activation in frontal regions, rather than reflecting task difficulty [20]. A recent meta-analysis [80] on healthy aging provided support for the PASA phenomenon; however, others have challenged it [81]. In addition to intra- and interhemispheric reorganization phenomena, we have the "cognitive reserve" hypothesis, which attributes successful cognitive processing in aging to complex interactions between genetic and environmental factors that influence brain reserve and the brain's ability to compensate for age-related pathologies [82]. Cognitive reserve is proposed to depend on both neural reserve and neural compensation, a distinction reflecting interindividual variability in the capacity to use resources efficiently, flexibly or differently while performing cognitive tasks but also to use alternative strategies in pathological situations. Accordingly, older adults can adapt to aging and flexibly cope with increased task demands by activating similar regions to younger adults, alternative regions, or both.

Alternatively, neurofunctional reorganization phenomena may be attributed to reduced neural efficiency, also known as dedifferentiation, resulting in reduced performance in older adults [31–34, 83]. According to the dedifferentiation hypothesis, aging reduces the specialization of neurons, which is critical for their optimal functioning [31]. Accordingly, increased

activation could be the result of randomly recruiting neurons in an attempt to meet processing demands [19], or could reflect the brain's failure to selectively recruit specific regions [34], whereas increasing task demands may aggravate the non-specificity of neural responses [84]. Evidence exists supporting the idea that neural responses are less specific in older adults than in younger ones, as demonstrated in the ventral visual cortex during a picture viewing task [33, 85], during a working memory task [86] (for a review, see [87]), and in motor evoked potentials [88]. It is not clear, however, whether this loss of neural specificity is the result of aging or could be attributed to older adults' greater experience in recognizing objects [33]. Nevertheless, the authors consider the latter explanation unlikely, since longer experience is expected to enhance rather than compromise the selectivity of neural responses. At the same time, it is thought that both compensation and dedifferentiation phenomena can occur in the same person simultaneously in different regions [86]. The dedifferentiation account would predict a reduction in performance together with an increase in activation, resembling the descending part of the inverted U-shaped relation between task demands and fMRI activation, according to CRUNCH.

Another potential explanation of age-related functional reorganization is that aging selectively affects the DMN. This network is normally activated when a person is not involved in any task but instead is monitoring their internal and external environment [2] and deactivated when the person performs cognitive tasks so as to reallocate attentional resources toward those tasks [35]. It is thought that the semantic network is largely activated at rest, as individuals are engaged in language-supported thinking when not performing specific tasks [89]. When a task is cognitively demanding, DMN deactivation has been found to be smaller and slower in older adults, implying that they become more easily distracted since their capacity to inhibit irrelevant information is compromised [28, 35, 90]. This result is in line with the inhibitory control view [91] and the cognitive theory of aging [2]. In difficult semantic tasks, maintained performance has been associated with increased segregation between DMN and semantic control regions at rest, whereas reduced performance was associated with increased verbal thinking at rest [92]. It is possible that aging reduces the efficiency with which attention is shifted away from resting areas toward task requirements, probably affecting the balance between DMN and task-related activity and resulting in reduced cognitive performance [2].

The neurofunctional reorganization proposals discussed above seem to be mutually exclusive as they tend to focus on and attribute meaningfulness to increased or decreased activation in isolated brain regions; however, none of them seems to fully capture and explain age-related reorganization [75]. Several researchers have attempted to identify the "common factor" [93] in age-related brain activation patterns to explain reorganization. Cabeza [19] considers that functional reorganization is likely to be unintentional and neuron-originated rather than reflecting a planned change in cognitive strategies, since it is manifested in simple tasks or following unilateral brain damage, over which people have little control. Conversely, Reuter-Lorenz and Cappell [28] consider it unlikely that such huge variability in brain activation stems from the same "common factor" or is due to age-related structural changes in the brain, because then it would be consistent across all tasks. Instead, aging seems to selectively affect specific regions, mainly default-mode regions and the dorsolateral PFC [2], whereas interindividual variability should be emphasized when accounting for age-affected cognitive domains [94].

Recent studies tend to combine data on functional, structural and lifetime environmental factors to explain reorganization in a more integrated way. Along these lines, the more comprehensive Scaffolding Theory on Aging and Cognition hypothesis proposes that aging is not characterized by an uncontrollable decline in cognitive abilities because the brain develops its own resilience, repairs its deficiencies and protects its functions [28, 95]. This idea is reflected

in aging models that emphasize the plasticity of the brain due, among other factors, to training interventions and their impact on neural structure, as well as functional and behavioral outcomes [96–98]. Short-term practice and lifelong learning impact younger and older adults differently [69]. Accordingly, engaging in intellectually challenging activities and learning throughout the lifetime but also in the shorter term could stimulate brain plasticity. The brain's capacity to resolve the mismatch between intellectual demands and available neurofunctional resources and to trigger behavioral adaptive strategies defines its plasticity and affects its knowledge systems and processing efficiency [69]. Plasticity is demonstrated in increased functional activation, especially in regions that are most structurally affected by aging because of atrophy, loss of gray and white matter density and cortical thinning, such as the frontoparietal network [97]. Aging could therefore be characterized by structural loss but also neural and functional adaptation to this loss, including through the use of new strategies [97]. Indeed, age-related overactivation seems to be a reliable, consistent pattern observed in multiple domains; overactivation may be localized, contralateral or seen in the frontoparietal multiple-demand network [99]. In summary, the more adaptable and dynamic the brain is, the better it maintains its cognitive abilities [100].

Specifically concerning the preservation of semantic memory in aging, it is not clear what mechanisms account for this preservation, which may be supported by the intersection of domain-general and linguistic abilities [66]. The literature presents varied findings about the adoption of neurofunctional activation patterns during semantic processing in aging. Two additional compensation-focused hypotheses have been proposed: the executive hypothesis refers to the recruitment of domain-general executive processes, evidenced in overactivation in the prefrontal, inferior frontal and inferior parietal brain regions to compensate for age-related cognitive decline [1, 101]; this process may be seen, for example, in performance on a semantic judgment task [56]. A meta-analysis of semantic memory studies that compared activation likelihood estimation between younger and older participants [22] found a shift in activation from semantic-specific regions to more domain-general ones, in line with the executive hypothesis.

The semantic hypothesis, on the other hand, also known as the left anterior-posterior aging effect (LAPA), refers to the recruitment of additional semantic processes in older adults, demonstrated by overactivation in "language" regions in the left posterior temporoparietal cortex [102, 103]. The greater decline in executive over language functions in older adults could justify this hypothesis: language is better maintained than executive processes [104]. Evidence for the semantic hypothesis was found in a study using a semantic judgment task where participants had to decide if a word denoted an animal or not. Older participants had more bilateral parietal, temporal and left fusiform activation than younger ones, who presented more dorsolateral activation; the authors interpreted this finding as showing that older participants rely more on semantic processes and younger ones rely more on executive strategies [105]. However, language and executive functions are very intertwined given that regions such as the left inferior frontal gyrus and the PFC are claimed to serve both kinds of functions; thus, the distinction between the semantic and executive hypotheses is a blurry one [53].

An alternative approach is taken by the good enough theory, which claims that participants tend to construct semantic representations that are "good enough," or shallow, rather than more complete or detailed ones, with the aim of performing the task at hand with the least possible effort and saving on processing resources [106–108]. This theory refers to language processing as a whole, but it can also be applied to the semantic representation of words, as inferred by the semantic judgment task used in our study. In this view, when participants, and especially older adults, face increased task demands, they may resort to a more "shallow" or superficial interpretation of the semantic judgment task they are required to perform. Thus,

instead of thoroughly analyzing all the semantic aspects of the words they are presented with (e.g., semantic features of an apple in comparison with a grape or cherry), may bypass some of them and make a quick decision. This kind of shallow processing might be manifested in decreased activation overall, as well as in the semantic control network, which is in charge of selectively controlling for certain semantic features while ignoring others [56]. This alternative explanation is in line with the idea that, at peak levels of demand, participants may become frustrated if they make many errors or have difficulty resolving competing representations, and may therefore deploy inefficient strategies [109].

In summary, some inconsistencies are found in the interpretation of results, as both increased and decreased activation are reported as the result of aging [2, 110]. Neurofunctional reorganization can take the form of both inter- and intrahemispheric changes in activation and manifests as both increased and decreased activation of specific regions [2]. When performance is compromised, reduced activation is interpreted as impairment, attributed to neural decline, inefficient inhibitory control or dedifferentiation [28]. On the other hand, when performance is maintained, it is claimed to be compensatory. Most studies seem to agree that activation increases and interpret it as a positive, compensatory phenomenon, whether it is understood as indicating increased attention or suppression of distracting elements [111]. Overactivation is also found in Alzheimer's disease and mild cognitive impairment patients, demonstrating either its compensatory role or its status as a progressive pathology predicting further decline [34, 35]. Neurofunctional reorganization of the aging brain is a complex process and further research is still required to enable us to describe a pattern of activation that integrates the existing findings into a comprehensive model that can be applied to semantic memory, one of the best-preserved cognitive fields in aging.

## Current study

The aim of this study was to identify whether aging affects the brain activity subserving semantic memory in accordance with the CRUNCH predictions, using a semantic judgment task with two levels of demands (low and high). Task demands were manipulated through semantic distance, which is found to influence both performance and brain activation levels [49, 61, 67, 112–115]. We hypothesized that brain activity and behavioral performance (dependent variables) would support the CRUNCH model predictions when demands on semantic memory were manipulated in younger and older adults (age and task demands: independent variables). More specifically, we expected (1) that the effects of semantic distance (low- versus high-demand relations) on neurofunctional activation and behavioral performance (accuracy and RTs) during the semantic judgment task would differ significantly between younger and older participant groups, and that younger adults would perform faster and more accurately than older adults. Furthermore, we predicted age group differences in brain activation in semantic control regions bilaterally that would be sensitive to increasing task demands [24]. This would be evident as a significant interaction effect between age group and task demands within regions of interest consisting of the core semantic control regions: IFG, pMTG, pITG and dmPFC. Such a finding would support the idea that the brain's ability to respond to increasing task demands declines with advancing age.

If such an interaction was not found between task demands and age, we expected the following results: (2) In the low-demand condition, both younger and older participants would perform equally well in terms of accuracy and make fewer errors than in the high-demand condition. However, we anticipated that older adults would present longer RTs and significant increases in activation in left-lateralized semantic control regions than younger participants. (3) In the high-demand condition, we expected that younger adults would perform better

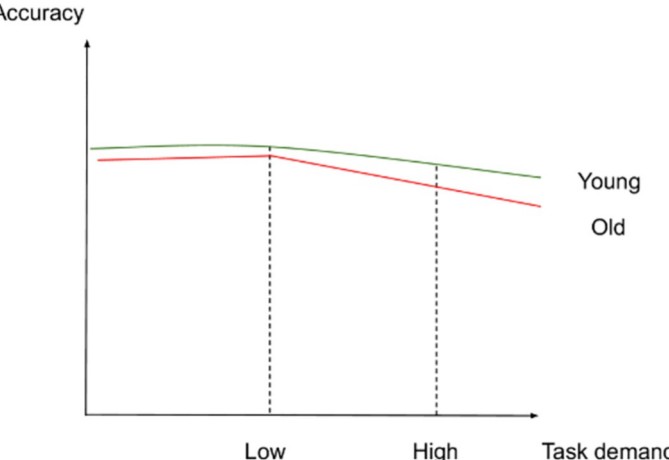

**Fig 1. Accuracy and task demands in younger and older adults.**

(higher accuracy and lower RTs) and present more activation in the left-hemisphere semantic control regions than older adults. Older adults were expected to perform less well than younger adults (lower accuracy and higher RTs) and to exhibit reduced activation in left-lateralized semantic control regions and increased activation in right-lateralized semantic control regions compared to the younger adults. Figs 1–3 illustrate the hypothesized relations between task demands and accuracy, RTs and activation in younger and older adults. The proposed theoretical relations between task demands and activation are represented in the decrease in activation in the left hemisphere (crossover interaction, Fig 3) and the increase in activation in the right hemisphere (difference in slopes interaction, Fig 4), supporting the CRUNCH-based predictions. These portray the main effects of age and task demands as well as their interactions, which are highlighted by thick lines.

These analyses look for the effects of age and task demands on task performance and on brain activation in separate analyses. Follow-up exploratory analyses within the ROIs explicitly

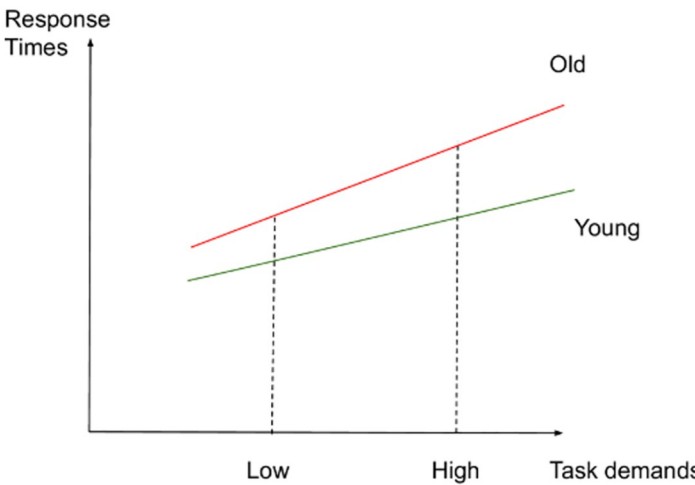

**Fig 2. RTs and task demands in younger and older adults.**

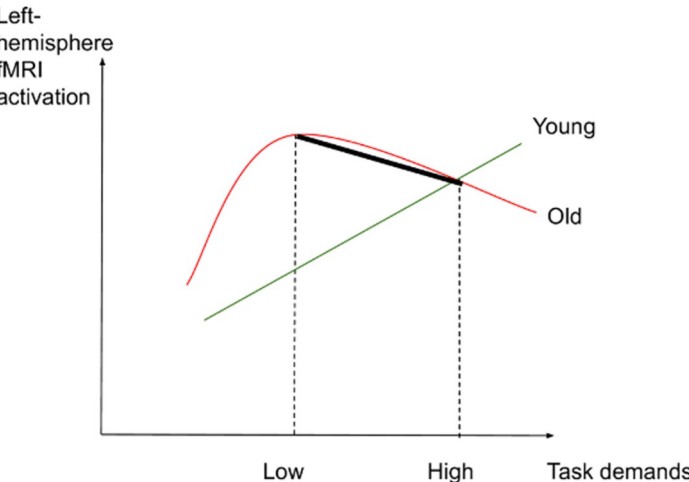

**Fig 3. Left-hemisphere activation and task demands in younger and older adults.**

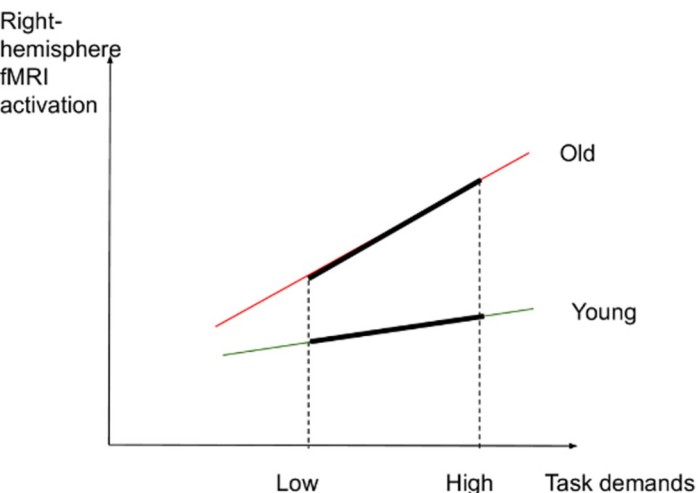

**Fig 4. Right-hemisphere activation and task demands in younger and older adults.**

test how differential brain activation is related to task performance. We hypothesized that older adults who had high levels of brain activation in left-lateralized semantic control regions during the high-demand condition, similar to the younger adults, would have higher levels of task performance (fewer errors and shorter RTs) than their counterparts whose brain activation was lower in these regions, as per the CRUNCH model, indicating that they had not yet reached their "crunch" point after which performance and activation decline. To support these hypotheses, at least one of the ROI mentioned was expected to be activated.

A control condition was part of the task and was designed to maximize perceptual processing and minimize semantic processing requirements [116, 117]. As a test of positive control, within-group comparisons with the control condition participants were expected to show activation in the primary visual and motor cortices, which are involved with viewing stimuli, response preparedness and motor responses [64, 118, 119]. No CRUNCH effects were expected in the control condition. Task effects within each age group were also tested and

activation was expected to be greater in the high- versus low-demand condition in both younger and older age groups.

This task design utilizes explicit definitions of low and high levels of task demand. However, each individual participant experienced their own subjective level of task difficulty. The perceived difficulty of triads was measured on a 7-point Likert scale (1: very easy, 7: very difficult). Subsequent analyses explored this question with heterogeneous slope models using individualized rescaled levels of task difficulty and compared brain activation with performance, brain activation with perceived difficulty and performance with perceived difficulty. This approach was developed to determine how the relationship between individual task difficulty and brain activity is affected by age group.

## Materials and methods

The authors complied with the requirements of the Centre de Recherche Institut Universitaire de Gériatrie de Montréal (CRIUGM) ethics committee and the Centre intégré universitaire de santé et de services sociaux du Centre-Sud-de-l'Île-de-Montréal (CÉR-VN: Comité d'Éthique de la Recherche—Vieillissement et Neuroimagerie), in line with the principles expressed in the Declaration of Helsinki. The CRIUGM ethics committee and the CÉR-VN approved this study (CÉR-VN registration number: 16-17-09). The approval letter is available in the Opens Science Framework (OSF) repository (DOI: 10.17605/OSF.IO/F2XW9).

For all methodological aspects of this study, compliance with the OHBM COBIDAS report/checklist [120] and guidelines [121] was aimed for as much as possible (full adherence, especially to the non-mandatory components, would have required extensive additional procedures, such as collecting information on participants' IQ). We have shared the preprocessed functional data sets in the MNI space publicly in the OSF (https://osf.io/) with a digital object identifier (DOI: 10.17605/OSF.IO/F2XW9) to permanently identify it [120], and we indexed it at the Canadian Open Neuroscience Platform (https://conp.ca/) to increase its findability. Data are organized following the Brain Imaging Data Structure (BIDS) to maximize shareability. Supporting documentation for this study is available at DOI: 10.17605/OSF.IO/F2XW9.

### Participants

A sample of 80 participants was required for this study: 40 in each group (Younger: 20–35 years old; Older: 60–75 years old; male = female). We contacted 265 participants in total (194 younger and 71 older ones). We recruited 84 participants (instead of the 86 initially planned), assuming that some would be excluded in the process due to poor task performance, excessive motion or technical issues. Recruitment took place from 17 June to 25 December 2021. Of the 84 participants, three (2 older males, 1 younger female) were excluded because data (either behavioral or imaging) were missing, and four (3 younger females and 1 older male) were excluded due to excessive motion. We ended up with 78 participants in total: 39 younger and 39 older adults.

Participants were recruited through the CRIUGM participant database and also through posters deployed in Montreal and on social media. Recruitment of participants followed the procedures for confidentiality of the CRIUGM. As such, participants were allocated with a code that did not allow for their identification. Only allowed researchers had access to identifying information of participants during data collection, and access to the document that linked code with participants' personal information was protected with a passcode. Participants were bilingual with French as their dominant language used on a daily basis; their second language might be English or another language. Multilingual participants were excluded, as speaking many languages may influence semantic performance [122]. Participants were matched for

education level, with college studies (CEGEP) as a minimum level, since education is a measure of cognitive reserve [82].

The initial inclusion criterion "born in Quebec" was dropped, as the exclusion criteria were deemed to hamper recruitment (the "born in Quebec" criterion had initially been adopted to account for linguistic differences between the French spoken in Quebec versus other francophone countries, including France). The COVID-19 pandemic made the recruitment and testing of participants particularly challenging, given the need to minimize older adults' exposure and the overbooking of the MRI machine at the Unité Neuroimagerie Fonctionnelle (UNF), which left few available slots for scanning and caused delays.

Participants underwent the following neuropsychological and health tests to determine their eligibility for the study as inclusion/exclusion criteria:

- A health questionnaire (prescreening was done by phone) to exclude participants with a history of dementia, drug addiction, major depression, stroke, aphasia, cardiovascular disease, diabetes, arterial hypertension or taking any drugs that could affect results. The prescreening included questions on bilingualism and use of French, which had be the dominant language (inclusion criteria) (the complete questionnaire is available on osf.io, DOI: 10.17605/OSF.IO/F2XW9).

- The Edinburgh Handedness Inventory scale: participants were right-handed with a minimum score for right-handedness of 80 [123].

- The MoCA (Montreal Cognitive Assessment) test, with a minimum cutoff score of 26 [124, 125].

- The MRI-compatibility checklist (UNF test) (available at https://unf-montreal.ca/forms-documents/).

- The following tests were also performed with participants:

- The Similarities (*Similitudes*) section of the Wechsler Adult Intelligence Scale (WAIS-III) test [126, 127].

- The Pyramids and Palm Trees Test (PPTT) (picture version) [128], which was used as a measure of semantic performance.

- The Habitudes de Lecture (Reading Habits) questionnaire (based on [129]) as a measure of cognitive reserve [82].

Participants provided written informed consent and were financially compensated for their participation according to the CRIUGM and Ethics Committee policies.

Due to practical problems resulting from COVID-19, minor adjustments in the timing of sessions were made in the original protocol. Table 1 sets out participants' demographic and neuropsychological characteristics.

## Power analysis

This sample size was based on a power calculation resulting from an age group comparison on a similar semantic task [130]. That data set used a Boston naming semantic task and compared healthy younger and older age groups. Based on that data set, effect size estimates were calculated from the contrasts for high versus low task demands within and between age groups. Effect sizes were extracted from the primary ROIs for this study as defined by a recent meta-analysis of semantic control [24]. From the identified locations, a 10 mm cube was defined to identify the effect size at the published location, the mean effect size and the robust maximum

**Table 1. Participants' demographic and neuropsychological scores.**

|  | Younger | Older |
|---|---|---|
| **Mean age** | 23.9 | 66.7 |
| **Sex (M/F)** | 17/22 | 14/25 |
| **Mean years of education** | 17.3 | 17.2 |
| **Mean frequency of use of French***  | 3.3 | 3.4 |
| **Mean frequency of use of English** | 2.15 | 1.07 |
| **Mean WAIS-III score (/33)** | 17 | 17.4 |
| **Mean PPTT score (/52)** | 49.8 | 50.9 |
| **Reading habits at 6 years old*** * | 4.1 | 3.1 |
| **Reading habits at 12 years old** | 3.3 | 3 |
| **Reading habits at 18 years old** | 3.3 | 3.5 |
| **Reading habits at 40 years old** | N/A | 3.4 |
| **Reading habits currently** | 3.1 | 3.5 |

* Frequency of use of French/English: 5: everyday; 4: several times per week; 3: several times per month; 2: several times per year; 1: never.

**Reading habits: 0: never; 1: once a year or less; 2: several times a year; 3: several times a month; 4: once or twice a week; 5: every day.

effect size in the ROI. Statistical power was then estimated using the G*Power tool [131]. Within-group measures had robust effect sizes and demonstrated that sufficient power (alpha = 0.05, beta = 0.90) was achieved with a sample size of 40 participants in each group. The between-group comparison of differential activation had sufficient power within the bilateral temporal gyri and medial PFC. In addition, the proposed study used more than twice as many trials as the data set used for power estimations. This would decrease the within-participant variation and increase the power above that provided by the earlier study [130]. The table of effect sizes used for the power analyses for within- and between-group comparisons is included as supplementary material on OSF.

## Materials

Participants were given a semantic similarity judgment task in French that is suitable for the Quebec context; the task was developed for this study and is similar to the PPTT [128]. The task involved triads of words resembling a pyramid; participants had 4 seconds to judge which of the two words below (target or distractor) was more closely related to the word above (stimulus). Both target and distractor words were associated in a semantic relation with the stimulus word. Participants were therefore required to select which of the two competing words had a stronger semantic relationship with the stimulus word, as measured by the semantic distance between the stimulus and the distractor. Two types of triads were used: (1) low-demand (distant) relations: the more distant the semantic relation between stimulus and distractor, the less demanding it would be to select the correct target; and (2) high-demand (close) relations: the closer the semantic relation between stimulus and distractor, the more demanding it would be to select the correct target, as competition between the target and distractor would be higher [61].

The task (150 triads in total) had two experimental conditions– 120 triads comprising 60 low-demand and 60 high-demand semantic relations—and one control condition with 30 triads. For the control condition, the task was to indicate which of the two consonant strings, which were presented pseudo-randomly, was in the same case as the target string (e.g., DKVP:

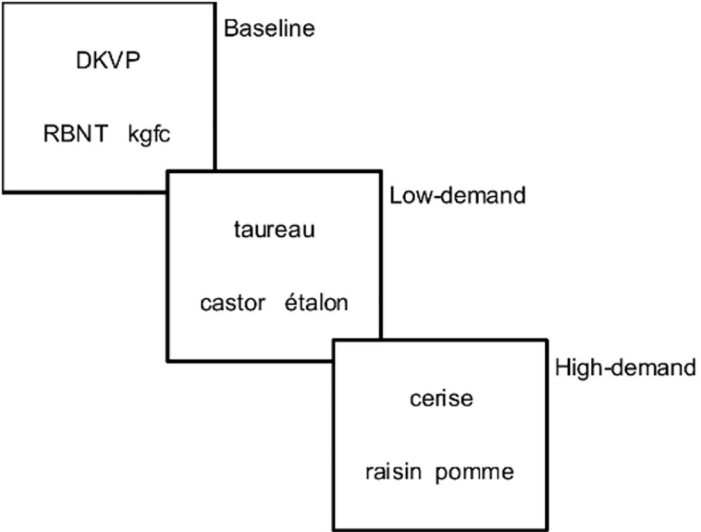

**Fig 5. Examples of triads.**

RBNT—kgfc). E-Prime automatically randomizes the location of correct responses (on the left or the right), to ensure variation. Fig 5 presents examples of stimuli for each condition.

## Stimuli development and pilot testing

The stimuli were developed as follows. In every condition, the targets and distractors were matched for type of semantic relation: taxonomic or thematic. Taxonomic relationships were developed as follows: the creation of taxonomic triads was based on (1) definition of semantic categories (e.g., farm animals, birds, tools, clothes, jewelry, wild animals, musical instruments, trees, insects, fruits, vegetables, fish, means of transportation, clothing, houses, furniture, other); (2) creation of triads based on the definitions below; (3) online testing of stimuli by 20 younger and 20 older pilot participants; testing of triads entailed choosing the correct answer in the triad and scoring the triad for difficulty on a 5-point Likert scale; (4) removal of triads that were inconsistent with the pilot participants' responses and scoring; (5) selection of the "most successful" triads (30 per condition), namely the triads to which 80% of pilot participants responded correctly and which were consistently scored as easy or difficult.

For thematic relations, the semantic distance was calculated with the help of a dictionary, the *Dictionnaire des associations verbales (sémantiques) du français* (http://dictaverf.nsu.ru/dict, version accessed in 2014), as a function of the number of respondents who associated two words (i.e., the larger the number of respondents, the more closely associated the two words are, and vice versa). Thus, a score of 1 means that only one person associated the two words (distant thematic relation) whereas a score of 100 means that 100 people associated them (close thematic relation). In addition, we calculated the words' frequency, based on the Lexique 3 database, which refers to films [132], and imageability, based on the Desrochers 3600 database [133]. Additional imageability ratings were collected from 30 French-speaking Quebecers for items without ratings in the database. A Pearson's correlation was performed between the scoring of this study's stimuli with scores for 30 test words from the Desrochers database to confirm that the ratings given for the new words were in line with those that already exist in the database. Participants' rating of items with a correlation value of less than 0.6 were excluded, as we considered that they were not concentrating on the task; 0.6 was chosen as

indicating extremely high power according to G*Power. The final imageability rating of an item was the mean of the scores given by all participants. ANOVAs and Bonferroni-corrected Tukey tests were performed to ensure that for every condition, the target and the distractor were matched for imageability and frequency. Finally, targets and distractors were matched for word length.

The stimuli were created by an iterative process, continuously testing and evaluating their appropriateness and aiming for an error rate of less than 40%. Pilot participants were used to test the stimuli, measure response times and gather comments. Each time, the four conditions were matched and an ANOVA was run on mean frequency, imageability and length.

Pilot participants included both younger and older francophone Quebecers (new participants each time). Testing involved the following steps: participants determined whether a triad was easy or difficult and rated it on a 5-point scale. Triads with a score of 3 were removed; then triads with scores of 1 and 2 were rated easy, while those with scores of 4 and 5 were rated difficult. Participants' answers needed to correspond with the definitions given (low- or high-demand). If they did not, the triad was removed. A score of 80% was defined as the cutoff for participants rating the triad according to the initial definition. Pilot participants also shared feedback about the duration of the task and whether they had enough time to respond.

After an initial set of triads was created, a team of linguists (headed by Phaedra Royle) evaluated it; their feedback and comments led to the replacement of 11 low-demand taxonomic triads, 10 high-demand taxonomic triads, 21 low-demand thematic triads, and 17 high-demand thematic triads. A first round of pilot testing took place with 6 younger participants and 5 older participants in January 2016. After triads were corrected or replaced, a second round of pilot testing pilot took place in February 2016 with 6 older participants and 3 younger participants. A third round of pilot testing with new stimuli took place in March 2016 with 6 new participants (3 younger and 3 older).

To evaluate the validity of the final stimuli for testing low versus high demand and younger versus older adults, a pilot evaluation of stimuli was conducted with 28 new participants (14 older adults, age range: 67–79 years old, female = 9; and 14 younger adults, age range: 21–35 years old, female = 10) with 60 triads (30 low-demand and 30 high-demand) using E-Prime. Repeated measures ANOVAs were done on mean accuracy and median response data for each level of task demand (control, low, high) across the two age groups. The results are described in the following sections.

**Accuracy.** The Greenhouse-Geisser estimate for the departure from sphericity was $\varepsilon = 0.63$. There was no significant interaction between age group and task demand, $F(1.27, 32.94) = 0.065$, $p = 0.85$, $\eta^2 = 0.0025$. The main effect of task demand was significant, $F(1.27, 32.94) = 10.36$, $p = 0.0015$, $\eta^2 = 0.28$. The estimated marginal means were Control = 0.84, Low = 0.80 and High = 0.72. The main effect of age group was not significant, $F(1, 26) = 0.34$, $p = 0.57$, $\eta^2 = 0.013$.

**Response times.** The Greenhouse-Geisser estimate for departure from sphericity was $\varepsilon = 0.54$. There was no significant interaction between age group and task demand, $F(1.08, 28.14) = 1.14$. $p = 0.30$, $\eta^2 = 0.042$. The main effect of task demand was significant, $F(1.08, 28.14) = 49.38$, $p < 0.0001$, $\eta^2 = 0.66$. The estimated marginal means were Control = 1,390 ms, Low = 2,230 ms and High = 2,292 ms. The main effect of age was significant, $F(1, 26) = 4.78$, $p = 0.038$, $\eta^2 = 0.15$.

Based on the above pilot data, we were able to confirm that our task impacted task performance (accuracy and RTs) differently in younger and older adults, in the expected directions.

The following definitions were used:

*Low-demand (distant) triads:*

- For taxonomic relations:

  All items (stimulus, target, distractor) belong in the same semantic category (e.g., animals). Stimulus and target words belong in the same semantic subcategory (e.g., birds). For example, *taureau*: *ÉTALON-castor* (bull: STALLION-beaver).

- For thematic relations:

  Both the target and distractor words are thematically related to the stimulus and belong in the list of answers provided by dictaverf. To ensure the largest possible distance, the target was the first adequate answer mentioned in dictaverf, whereas the distractor was the last or next-to-last answer, meaning that it had a score close to 1. For example, *sorcier*: *village-BAGUETTE* (wizard: village-WAND).

  Alternatively, when the above criterion could not be applied, the following criteria were adopted to ensure the largest distance possible: (1) when the distractor word had a score of 1 (which means only 1 person provided this answer), the target could have any score; (2) when the distractor word had a score between 2 and 5, the target word had to have a score above 10; and (3) when the difference between the target and distractor words was greater than 100, then the actual scores did not matter.

  *High-demand (close) triads:*

- For taxonomic relations:

  All items in the triad come from the same semantic subcategory (e.g., birds). The stimulus and target items share a visual or structural feature whereas the distractor word does not. For example, *cerise*: *RAISIN-pomme* (cherry: GRAPE-apple): cherries and grapes share a similar size and bunch structure.

- For thematic relations:

  Both the target and distractor words are thematically related to the stimulus. The target was the first appropriate answer mentioned in dictaverf whereas the distractor had a score lower than or equal to half of the target's score and greater than or equal to 4. The cutoff score was chosen empirically so that the distractor's score was always larger for high-demand rather than low-demand triads. This criterion was used to ensure that the distractor was a more frequently mentioned answer but distant enough from the target (e.g., half of the people mentioned the distractor instead of the target). For example, *enfant*: *JOUET-sourire* (child: TOY-smile).

## Experimental design

**Deviations from the registered report protocol.** The current study methodology is based on the previously published registered report protocol [134]. Several deviations were made from the original registered report protocol published in 2021.

- The fMRI acquisition parameters were tweaked. The team proposed minor adjustments to the technical platform at the UNF at the CRIUGM, *prior to data collection*, which were communicated by email to the *PLOS One* editors. These adjustments represent minor technical precisions motivated by recent technological improvements to the MRI platform at the research institute, and do not affect the preregistration process. The new acquisition parameters were uploaded to the OSF repository, in both track changes mode and the final version (SSTA_20210621).

- Session no. 3 (subjective evaluation of task difficulty) took place immediately after the fMRI acquisitions, and not one week later as originally planned, as it would have been a burden for participants to return to the CRIUGM and posed risks of loss to follow-up.

- The "born in Quebec" inclusion criterion was dropped, as it was deemed too strict for recruitment. As explained above, COVID-19 made the recruitment and testing of participants particularly challenging, given the need to minimize older adults' exposure and the overbooking of the UNF's MRI machine.

- Ultimately, 84 participants were recruited, instead of 86, as initially planned, since COVID-19 restrictions hampered recruitment.

- After 13 participants were scanned, the laptop where E-Prime was installed crashed (29 October 2021); consequently, we used the UNF PC on which E-Prime 3.0 was installed.

- Due to a problem with field map correction (black images) with seven participants, this step was not performed.

- The first participant to be scanned commented about fatigue in the scanner; as a result, the order of scanning was reversed so that the task-based fMRI was collected before the other non-cognitive imaging was done (in OSF.io, SSTA.pdf supersedes neuro.pdf).

**Session 1: Neuropsychological tests.** Participants were recruited from the CRIUGM pool of participants and with public announcements; their eligibility was initially assessed during a phone interview (health questionnaire, French language, right-handedness and MRI compatibility). If eligible, the participant participated in the first experimental session (approximately 45 minutes), during which they signed the informed consent and MRI-compatibility forms, completed neuropsychological tests (see Participants section above) and trained with 15 practice triads (5 for each condition). Participants who qualified for the fMRI scanning session (met the inclusion criteria from the health questionnaire, MRI-compatibility questionnaire, MoCA and Edinburgh Handedness Inventory scale) took the second session one week later (maximum two weeks later).

**Session 2: fMRI scanning.** For the second experimental session, the participants' time commitment was 90 minutes to allow for practice with the triads, getting ready and leaving, and following the COVID-19 requirements. During this session, participants listened to task instructions, and practiced with three triads (1 per condition). Participants' vision was corrected, if necessary, with MRI-175 compatible lenses, according to their prescription shared at the previous session. In addition, pregnancy tests were carried out when relevant (as per the UNF's requirements), earplugs were provided to reduce machine noise and participants were instructed to remain still in the scanner (foam rubber pads in the head coil also restricted movement). Participants then started the actual task in the scanner. Stimuli were presented with E-Prime version 2.0.10.356 software run on Microsoft Windows 10 (for the first 13 participants), and E-Prime 3.0 for the remaining ones. An LCD projector projected the stimuli to a mirror above the participant's head. Participants selected their responses by pressing buttons using the index fingers of both hands on the MRI-compatible response box. A response on the right was made with the right hand and a response on the left with the left hand. Response data and RTs were recorded with E-Prime for further analysis. No feedback was given to participants. Participant testing alternated between younger and older adults to minimize any bias due to scanner changes or upgrades.

The semantic task was event-related. Triads were presented for 4 s, during which participants responded. A black screen followed for approximately 2.2 s (this interstimulus interval

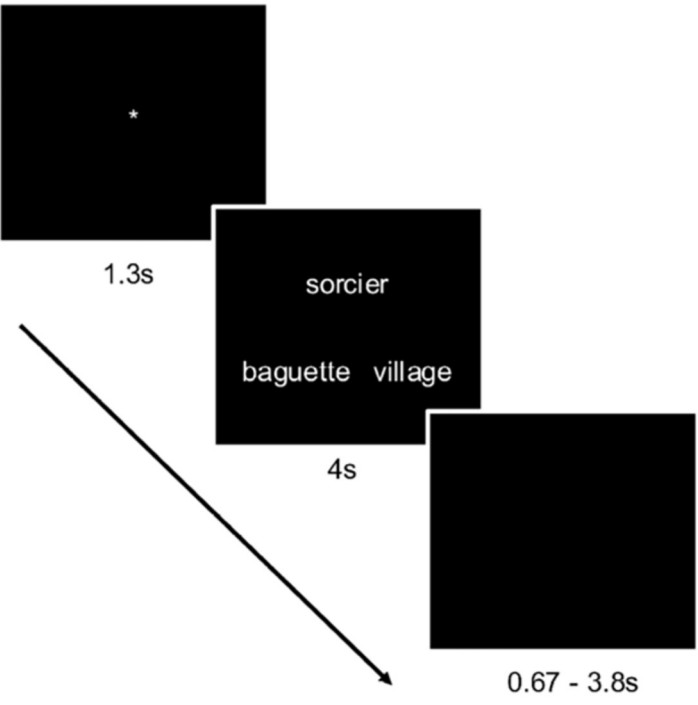

**Fig 6. Flow of a sample trial.**

(ISI) varied randomly between 0.67 s and 3.8 s to maximize variance in the BOLD signal and ensure unpredictability). A fixation point appeared for 1.3 s to prepare the participant for the next trial. The whole trial lasted between 5.97 s and 9.10 s, with a mean of 7.5 s. See below for a description of the methods used to determine the ISIs. Black screens were presented at the beginning and the end of the runs. Information on the scanning flow is presented in Fig 6.

The task was split into two runs with 75 triads per run (30 low-demand, 30 high-demand and 15 control triads), interspersed in a pseudo-random fashion, so that no more than four trials of the same condition or type were repeated in a row. The duration of each run was 9:45 minutes. The whole session lasted approximately 45 minutes, including a 5-minute break between runs 1 and 2.

**Session 3.** To address perceived task difficulty, an additional session was held with participants immediately following the fMRI acquisitions, at which they rated each triad for difficulty using a 7-point Likert scale (1: very easy, 7: very difficult). The objective of this session was to further assess whether perceived difficulty correlated with actual performance scores (accuracy rates and RTs) and with levels of activation in the younger and older adults. In other words, we investigated whether greater perceived difficulty correlated with increased RTs and reduced accuracy, as well as levels of activation in semantic control regions.

**Stimulus order and interstimulus intervals.** To maximize design efficiency, optimal trial ordering and ISIs were chosen [135]. The methodology used simulated random ordering of the three conditions. In addition, the ISIs were randomly drawn from gamma distributions across a range of parameter values (shape: 0.1 to 10, scale: 0.1 to 5). This approach included expected error rates derived during the stimulus pilot tests. A total of 800,000 simulations were performed. The ISI distribution and specific list were chosen, as was the condition order in

which there was the smallest decrease in required BOLD signal response for detection as errors increased. The related ISIs have been uploaded to the OSF platform.

**fMRI data acquisition.** Prior to data collection, minor deviations were applied to the fMRI acquisition parameters, which were also uploaded on the OSF platform and of which the journal editor was informed. These adjustments represent minor technical precisions motivated by recent technological improvements to the MRI platform at our research institute and as recommended by the team responsible for the technical platform at the UNF.

Functional scans were performed on a 3Tesla Syngo MR E11 Prisma_fit Siemens MRI machine with 32 channel receive-coil at the UNF, CRIUGM. The start of the stimulus presentation software was triggered by a pulse sent from the MRI to the laptop. To detect effects between conditions and ensure a good fMRI signal in the brain, pilot data collected using the scanning protocol described here suggested a minimum temporal signal to noise ratio (TSNR) of 20 throughout the brain [136]. A participant's data were excluded if the TSNR, assessed from that participant's time series, was below 20. We acquired T1-weighted MRI images for coregistration with fMRI data and atlases and to identify ROIs to be used as masks in the functional data analysis. An meMPRAGE (multi-echo MPRAGE) sequence (704 total MRI files) was acquired with 1 x 1 x 1 mm resolution, 2.2 s repetition time, 256 x 256 acquisition matrix, a Field of View (FOV) of 256 mm covering the whole head and echo times of 1.87 ms, 4.11 ms, 6.35 ms, and 8.59 ms, for a total of 704 meMPRAGE slices (176 slices x 4 echoes = 704 slices). The phase encoding direction was A-P (anterior to posterior) and superior-inferior, and slice orientation was sagittal with a flip angle of 8 degrees. Following scanning of the first participant and their comment about fatigue, we reversed the order so that the task-based fMRI was done before the resting state and arterial spin labeling.

For the functional scans (runs 1 and 2), T2*-weighted BOLD data were acquired on the entire brain (including the cerebellum) using an Echo Planar Imaging (EPI) sequence with 50 slices, resolution 2.5 x 2.5 x 3 mm, echo time of 20 ms, repetition time of 3 s and a flip angle of 90 degrees and parallel imaging (R = 2). Field of view was 220 x 220 mm and the acquisition matrix was 88 x 88, in the AC-PC direction minus approximately 20 degrees covering 150 mm in the *z*-direction. Slice order was ascending-interleaved. For each run, 195 volumes were collected. Functional images were reconstructed to the collected matrix size with no prospective motion correction. Two initial dummy scans were collected and discarded by the MRI, allowing for T1 saturation.

The Siemens default gradient field map sequence for field map distortion correction was acquired after each sequence for inhomogeneity correction; 50 axial slices were acquired, with resolution of 2.5 x 2.5 x 3 mm, repetition time of 520 ms, and echo times of 4.92 ms and 7.38 ms. Phase-encode direction was A-P in the same axial orientation and same angulation as the EPI sequence.

## Analysis methodology

**Behavioral analysis methods.** Repeated measures analysis tested for age group and task demand-related effects on task accuracy and RTs while controlling for sex. The Greenhouse-Geisser estimate corrected for any departure from sphericity and Tukey post-hoc tests were Bonferroni corrected. Analyses were done with Jamovi software (The Jamovi project, 2021) (Version 1.6), retrieved from https://www.jamovi.org. Any missing or incomplete data for a participant meant that the participant was excluded.

**Image preprocessing methods.** Preprocessing image analysis was performed with SPM12 software on the Narval cluster of Calcul Québec at Compute Canada. Images were corrected for slice timing (differences in slice acquisition time), with ascending-interleaved slice order

and using the acquisition time for the middle slice as the reference. We used field map correction to correct EPI images for distortion using the Calculate VDM toolbox and the first EPI image as a reference. The gradient field map images were presubtracted by the scanner to provide phase and magnitude data separately. Motion correction was applied for within-subject registration and unwarping. Motion parameters were used later as confound variables. Three participants with acute motion parameters of more than 2 mm, or 1-degree rotation, between scans in any direction were excluded. Four additional participants were excluded as they had missing data (i.e., less volumes). EPI functional volumes were registered to the average anatomical volume calculated by the machine over the four echoes of the meMPRAGE T1-weighted anatomical scan. The mean anatomical image was used as the reference image and for quality control. Anatomical variations between participants were reduced by aligning the structural images to the standard space MNI template, followed by visual inspection of their overlay. Data were visually inspected for excessive motion by inspecting with SPM's check reg tool to make sure that five landmarks across the EPI brain were well aligned with the standard space brain. Seven participants produced black images during this process. A problem was deemed to exist with their field map correction, and so field map correction was removed. An 8 mm full width at half maximum (FWHM) Gaussian blur was then applied to smooth images within each run. The final voxel size after preprocessing was 3 x 3 x 3 mm.

**fMRI data analysis methods.** fMRI data analysis was performed with SPM12 focusing on primary ROIs for semantic control. Using files created by E-Prime during stimulus presentation, stimulus onset files were created and regressors were defined. For the first-level (intrasubject) analysis, a General Linear Model (GLM) employing the canonical Hemodynamic Response (HDR) Function and its derivative, both convolved with a model of the trials, was used to estimate BOLD activation for each subject as a function of condition for the fMRI task. The inclusion of the derivative term accounted for interindividual variations in the shape of the HDR. Correct trials were modeled separately for low- and high-demand conditions. Incorrect trials for low and high demands were modeled together in their own regressor and not further investigated. Each participant's fMRI time series (2 runs) were analyzed in separate design matrices using a voxel-wise GLM (first-level models). Movement parameters obtained during preprocessing, and their first and second derivatives, were included as covariates (regressors) of no interest to reduce the residual variance and the probability of movement-related artifacts. A high-pass filter with a temporal cutoff of 200 s and a first-order autoregressive function correcting for serial autocorrelations were applied to the data before assessing the models. Two contrasts of interest were calculated, collapsing across the two runs. These contrasts were low-demand, correct trials > control and high-demand, correct trials > control. These contrasts were used for second-level group analyses to compare the effects of age group and task demand.

The analysis first tested for an interaction between age group and task demands. A significant finding would support hypothesis 1. If such an interaction was not found, main effects would be further tested. Support for hypothesis 2 would mean that the older age group had significantly greater activation than the younger age group in left semantic control regions. Support for hypothesis 3 would mean that, in the high-demand condition, the younger age group would present significantly greater activation than the older group in the left semantic control regions; it would also mean that the older age group had significantly greater activation than the younger group in the right semantic control regions.

To account for differences in HDR between younger and older adults, the event-related first-level statistical model of the fMRI data included the event-chain convolved with the double-gamma HDR function and its first derivative. The inclusion of this extra regressor would

capture variance in the data due to any interparticipant or intergroup variations in the shape of the HDR.

**Defining the anatomical/functional ROIs.** This study's hypotheses depend on ROIs that include semantic control regions associated with low- and high-demand conditions. To identify ROIs of the semantic control network demonstrating demand-related differences in brain activation, the results of a recent meta-analysis were used [24]. This analysis utilized data from 126 comparisons and 925 activation peaks and is the most comprehensive and up-to-date meta-analysis of semantic control networks. The results identified 20 highly significant peak locations throughout the inferior frontal gyrus, insula, orbitofrontal cortex, precentral gyrus, middle and inferior temporal gyri and fusiform gyrus; see Table 2 for specific $x,y,z$ locations. Spheres 10 mm in diameter were created at each of these locations and the corresponding contralateral locations, by flipping the sign of the $x$-coordinate. Participant level parameter estimates (contrast values) were extracted using MarsBar [137]. This approach applied the methods presented in a recent analysis of the CRUNCH effect in a similar population [40]. Correction for multiple comparisons at alpha < 0.05 used the false discovery rate across the 40 ROIs [138]. Secondary, exploratory analyses of the more general semantic control network used the maps of semantic control for domain-general control, as identified in the meta-analysis in [24].

**Region of interest analysis methods.** Spheres 10 mm in radius were drawn around each of the 20 $x,y,z$ coordinates provided in Jackson's 2021 semantic control meta-analysis [24] (Fig 7). For each participant, the mean brain activity within each ROI for both task demand levels

**Table 2. Semantic control activation likelihood.**

| Cluster number | Region of activation | Max ALE value | Z value | Peak MNI coordinate | | |
|---|---|---|---|---|---|---|
| | | | | X | Y | Z |
| 1 | Left IFG (pars triangularis, orbitalis & opercularis), insula, OFC & precentral gyrus | 0.093 | 10.679 | −48 | 22 | 20 |
| | | 0.060 | 7.809 | −46 | 24 | −2 |
| | | 0.055 | 7.402 | −50 | 30 | 0 |
| | | 0.037 | 5.495 | −34 | 26 | −6 |
| | | 0.036 | 5.378 | −46 | 40 | −10 |
| | | 0.034 | 5.134 | −48 | 34 | −12 |
| | | 0.029 | 4.565 | −30 | 24 | −16 |
| | | 0.024 | 3.909 | −44 | 2 | 48 |
| | | 0.020 | 3.349 | −38 | 28 | −22 |
| 2 | Left pMTG, pITG & pFG | 0.039 | 5.725 | −54 | −42 | 4 |
| | | 0.037 | 5.500 | −46 | −48 | −16 |
| | | 0.037 | 5.478 | −46 | −56 | −12 |
| | | 0.036 | 5.386 | −56 | −46 | −4 |
| | | 0.021 | 3.514 | −50 | −68 | −2 |
| 3 | Bilateral dmPFC | 0.058 | 7.697 | −2 | 20 | 52 |
| | | 0.034 | 5.225 | 2 | 28 | 36 |
| | | 0.025 | 4.047 | −4 | 8 | 58 |
| 4 | Right IFG (pars orbitalis) & insula | 0.046 | 6.502 | 32 | 24 | −6 |
| | | 0.019 | 3.294 | 30 | 18 | −18 |
| 5 | Right IFG (pars triangularis) | 0.044 | 6.217 | 50 | 24 | 26 |

IFG = inferior frontal gyrus, OFC = orbitofrontal cortex, $p$ = posterior, ITG = inferior temporal gyrus, MTG = middle temporal gyrus, FG = fusiform gyrus, dmPFC = dorsomedial prefrontal cortex. This table was copied from Table 1 in [24] and refers to semantic control contrast peak coordinates.

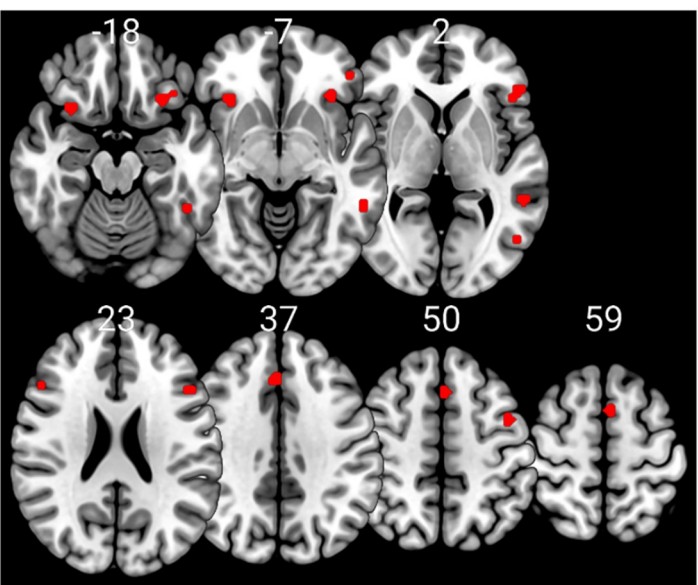

**Fig 7. Regions of interest derived from a recent meta-analysis of semantic control.** Numbers are axial slice locations in millimeters.

was calculated. This provided two numbers per person for each of the 20 ROIs. A linear mixed effects model, similar to that used for the behavioral analyses, was fit to each ROI using the lme4 package in r (v1.1–28) [139].

## Results

### Behavioral data analysis

As explained in the Participants section, this study included data from a total of 78 participants: 39 younger adults, with a mean (standard deviation, SD) age of 24.0 years (3.48) and a range of 19 to 32 years, and 38 older adults, with a mean (SD) age of 66.6 years (4.08) and a range of 60 to 74 years. There were 22 females and 18 males in the younger age group and 25 females and 13 males in the older age group. A $\chi^2$-test of independence confirmed no significance relationship between sex and age group counts ($\chi^2(1) = 0.947$, $p = 0.330$).

The results of the cognitive assessments are shown in Table 3.

Descriptive measures of task-related behavioral performance are shown in Table 4. In predicting response time, the main effects of age group ($F(1, 74) = 9.78$, $p = 0.00253$) and task demand ($F(1, 229) = 5.54$, $p = 0.0194$) were significant. The interaction between age group and task demand was not significant ($F(1, 229) = 0.00883$, $p = 0.925$) and the main effect of sex was also not significant ($F(1, 74) = 0.728$, $p = 0.396$). The main effect of age group was driven by longer RTs in older adults (mean (standard error, SE) = 2,265 (39.1) ms) than the younger adults (mean (SE) = 2,096 (37.8) ms) a difference of 169 ms. The main effect of task demand was driven by longer RTs in the high-demand condition (mean (SE) = 2,197 (28.3) ms) than in the low condition (mean (SE) = 2164 (28.3) ms), a difference of 33 ms. The random component of the model (participant, intercept) was significant (ICC = 0.774, $\chi^2(1) = 246$, $p < 0.0001$).

A Fisher logit transform of mean accuracy scores was used to predict accuracy, and a repeated-measures ANOVA model was applied. Thus, a minimal proportion of variation in

**Table 3. Results of cognitive assessment.**

|  | Younger mean (SD) | Older mean (SD) | *t* | *p* | Effect size |
|---|---|---|---|---|---|
| **MoCA** | 28.3 (1.15) | 28.4 (1.33) | 0.519 | 0.605 | 0.118 |
| **PPTT** | 49.8 (2.13) | 50.9 (1.22) | 2.71 | 0.00838 | 0.613 |
| **WAIS-III** | 17.1 (4.01) | 17.4 (3.43) | 0.409 | 0.684 | 0.0926 |

**Table 4. Task-related behavioral measures.**

|  | Response Time (ms) | | Accuracy | |
|---|---|---|---|---|
|  | **Low** | **High** | **Low** | **High** |
| **Young** | 2078 (265) | 2109 (228) | 0.86 (0.057) | 0.79 (0.064) |
| **Old** | 2240 (243) | 2274 (231) | 0.88 (0.043) | 0.78 (0.055) |

the data was attributed to between-participant differences. The main effect of task demand was significant ($F(1, 74) = 138$, $p < 0.0001$, $\eta^2 = 0.290$). This effect was driven by higher mean transformed accuracies in the low-demand condition (mean (SE) = 1.90 (0.0457)) than in the high-demand condition (mean (SE) = 1.30 (0.0394)). The interactions between task demand and sex ($F(1, 74) = 0.193$, $p = 0.622$, $\eta^2 = 0.00$) and between task demand and age group ($F(1, 74) = 2.62$, $p = 0.110$, $\eta^2 = 0.00549$) were not significant. The main effects of age group ($F(1, 74) = 0.0505$, $p = 0.823$, $\eta^2 = 0.00$) and sex ($F(1, 74) = 0.0236$, $p = 0.878$, $\eta^2 = 0.00$) were not significant.

## Results of fMRI analyses

Second-level analyses used a repeated-measures univariate analysis and tested for task demand effects, age effects, and their interaction. This analysis used the sandwich estimator, which appropriately accounts for the within-subject correlation in repeated-measures data [140]. Analyses used two images per participant. These were the first-level contrasts of the low-demand versus the control condition and the high-demand versus the control condition. Statistical significance at alpha < 0.05 was assessed using 1,000 resamples and threshold-free cluster enhancement and family-wise error corrections for multiple comparisons across voxels [141, 142].

There were no significant voxels for the test of the interaction between task demand and age. Of the two main effects, only the effect of age group demonstrated any significant voxels within the bilateral occipital cortex and the cerebellum. Despite the absence of task demand effects and minimal age effects, the experiment demonstrated robust task-related brain activity. Significant brain activity was evident across bilateral inferior frontal, parietal, supplementary motor, temporal, and occipital brain regions, see Fig 8. Tables 5–9 list the results for multiple local peaks of activity.

## Region of interest analyses

ROI analyses demonstrated only uncorrected significant effects of task demands within the right inferior frontal gyrus. The effect in the pars triangularis region was driven by lower activity at the high-demand level. Within the pars orbitalis, the effect was driven by greater negative direction activity at the high-demand level. Despite the lack of significant interactions or age effects, and the minimal task demand effects, there were strong task-related effects as

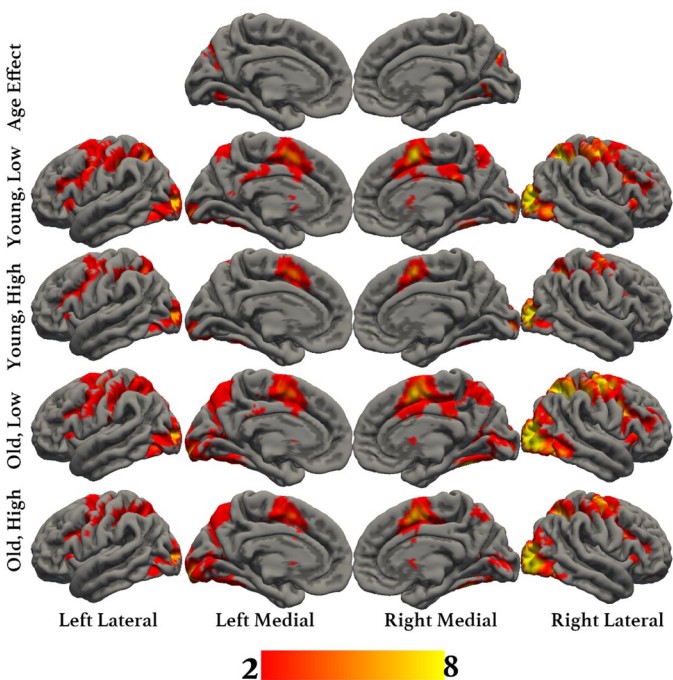

**Fig 8. Overlays of all results demonstrating significant brain activity using threshold-free cluster enhancement with family-wise error correction for multiple comparisons and 1,000 resamples.** Colors correspond to Z-values ranging from 2 to 8. The age effect represents increased brain activity for older compared to younger participants. The low results are for low task demands greater than the control condition. The high results are for high task demands greater than the control condition.

**Table 5. Main effect of age.**

| Region | Hemi. | B.A. | x | y | z | Z | k |
|---|---|---|---|---|---|---|---|
| Cerebellum, crus1 | R | — | 15 | −52 | −16 | 4.98 | 221 |
| Cerebellum, crus1 | L | — | −15 | −67 | −7 | 5.39 | 90 |
| Calcarine sulcus | L | 18 | −12 | −76 | 35 | 5.52 | 174 |
| Mid. occipital gyrus | R | 18 | 12 | −82 | 48 | 5.04 | 8 |

Hemi: hemisphere; B.A.: Brodmann area *k*: cluster size,—: a location with no representation within the Brodmann atlas. Results are corrected for multiple comparisons using threshold-free cluster enhancement, family-wise error correction and 1,000 resamples.

demonstrated by significantly greater than zero activity shown by the estimated marginal means. Table 10 shows the results for the ROI analyses.

## Timeout analyses

The number of timeouts, trials with no response, increased with task demands. This unanticipated observation was not included in the registered report making the following results exploratory. In predicting the number of time outs (delayed responses), the main effect of task demand was significant ($F(1,73) = 10.9$, $p = 0.00148$). The main effects of age group ($F(1,73) = 0.00572$, $p = 0.940$) and sex ($F(1,73) = 0.103$, $p = 0.750$) were not significant. The interaction between task demand and age group was not significant ($F(1,74) = 0.00084$, $p = 0.977$). The

**Table 6. Younger adults, low task demand.**

| Region | Hemi. | B.A. | x | y | z | Z | k |
|---|---|---|---|---|---|---|---|
| Lingual gyrus | R | 18 | −24 | −91 | −7 | 8.11 | 11838 |
| Inf. occipital gyrus | L | 18 | 24 | −91 | −4 | 7.81 | — |
| Inf. occipital gyrus | R | 19 | −30 | −85 | −7 | 7.77 | — |
| Mid. occipital gyrus | L | 19 | 33 | −88 | −1 | 7.70 | — |
| Sup. parietal cortex | L | 7 | 27 | −58 | 47 | 6.88 | — |
| Sup. parietal cortex | R | 7 | −27 | −58 | 53 | 6.78 | — |
| Sup. occipital gyrus | R | 7 | −24 | −67 | 35 | 6.72 | — |
| Supp. motor area | R | 6 | −6 | −1 | 53 | 6.67 | — |
| Supp. motor area | L | 6 | 6 | 8 | 53 | 6.54 | — |
| Supp. motor area | R | 6 | −6 | 8 | 50 | 6.51 | — |

Hemi: hemisphere; B.A.: Brodmann area; k: cluster size,—: a local maximum within a larger cluster. Results are corrected for multiple comparisons using threshold-free cluster enhancement, family-wise error correction and 1,000 resamples.

main effect of task demand was driven by more timeouts in the high-demand condition (mean (SE) = 1.58 (0.300) than in the low-demand condition (mean (SE) = 0.834 (0.300), a difference of 0.746 trials. The random component of the model (participant, intercept) was significant (ICC = 0.697, $\chi^2(1) = 49.0$, $p < 0.0001$).

**Table 7. Younger adults, high task demand.**

| Region | Hemi. | B.A. | x | y | z | Z | k |
|---|---|---|---|---|---|---|---|
| Lingual gyrus | R | 18 | −24 | −91 | −7 | 7.74 | 3656 |
| Inf. occipital gyrus | R | 19 | −30 | −85 | −7 | 7.24 | — |
| Supp. motor area | R | 6 | −6 | 8 | 50 | 6.37 | — |
| Sup. occipital gyrus | R | 7 | −24 | −67 | 35 | 6.11 | — |
| Precentral gyrus | R | 6 | −27 | −4 | 50 | 6.08 | — |
| Sup. parietal cortex | R | 7 | −24 | −61 | 50 | 6 | — |
| Supp. motor area | R | 6 | −6 | −1 | 53 | 5.98 | — |
| Precentral gyrus | R | 44 | −42 | 5 | 29 | 5.96 | — |
| Supp. motor area | L | 6 | 6 | 8 | 53 | 5.7 | — |
| Precentral gyrus | R | 6 | −54 | −1 | 44 | 5.53 | — |
| Inf. occipital gyrus | L | 18 | 24 | −91 | −4 | 7.39 | 2044 |
| Sup. parietal cortex | L | 7 | 27 | −58 | 47 | 5.62 | — |
| Precentral gyrus | L | — | 36 | −16 | 53 | 5.54 | — |
| Precentral gyrus | L | 6 | 39 | −13 | 65 | 5.05 | — |
| Precentral gyrus | L | 6 | 27 | −4 | 47 | 4.91 | — |
| Mid. occipital gyrus | L | 19 | 30 | −70 | 26 | 4.8 | — |
| Fusiform | L | 37 | 36 | −67 | −10 | 4.76 | — |
| Postcentral | L | — | 54 | −13 | 50 | 4.73 | — |
| Fusiform | L | 37 | 33 | −40 | −22 | 4.33 | — |
| Fusiform | L | 37 | 36 | −58 | −13 | 4.29 | — |
| Inf. frontal oper. | L | 44 | 42 | 8 | 29 | 4.71 | 80 |
| — | R | 47 | −30 | 26 | 2 | 4.09 | 18 |

Hemi: hemisphere; B.A.: Brodmann area; k: cluster size,—: a local maximum within a larger cluster. Results are corrected for multiple comparisons using threshold-free cluster enhancement, family-wise error correction and 1,000 resamples.

**Table 8. Older adults, low task demands.**

| Region | Hemi. | B.A. | x | y | z | Z | k |
|---|---|---|---|---|---|---|---|
| Inf. occipital gyrus | R | 18 | −27 | −88 | −4 | 8.05 | 17869 |
| Mid. occipital gyrus | L | 19 | 36 | −88 | −1 | 7.95 | — |
| Sup. occipital gyrus | R | 7 | −21 | −64 | 44 | 7.3 | — |
| Inf. parietal cortex | R | 40 | −33 | −46 | 44 | 7.28 | — |
| Precentral gyrus | L | — | 36 | −16 | 50 | 7.13 | — |
| Sup. occipital gyrus | R | 19 | −24 | −61 | 32 | 7.12 | — |
| Sup. parietal cortex | L | 7 | 24 | −61 | 47 | 6.92 | — |
| Precentral gyrus | L | 6 | 33 | −16 | 59 | 6.85 | — |
| Mid. occipital gyrus | L | 7 | 27 | −61 | 38 | 6.79 | — |
| Precentral gyrus | R | 6 | −39 | −7 | 59 | 6.69 | — |
| Sup. temporal gyrus | L | 42 | 54 | −43 | 20 | 3.01 | 12 |
| Supramarginal gyrus | L | — | 51 | −37 | 26 | 2.59 | — |
| Sup. temporal gyrus | L | — | 60 | −37 | 23 | 2.51 | 1 |

Hemi: hemisphere; B.A.: Brodmann area; $k$: cluster size,—: a local maximum within a larger cluster. Results are corrected for multiple comparisons using threshold-free cluster enhancement, family-wise error correction and 1,000 resamples.

## Analyses of perceived difficulty (Likert scales)

**Age and condition effects on perceived difficulty.** In predicting perceived difficulty scores using a 7-point Likert scale, the main effects of age group ($F(1, 73) = 12.2$, $p = 0.0008$) and task demand level ($F(1, 74) = 73.0$, $p < 0.0001$) were significant. The main effect of sex ($F(1, 73) = 0.876$, $p = 0.352$) was not significant. The interaction between perceived difficulty and age group was not significant ($F(1, 74) = 0.230$, $p = 0.633$). The main effect of perceived difficulty was driven by higher scores in the high-demand condition (mean (SE) = 2.55 (0.081)) than in the low-demand condition (mean (SE) = 2.23 (0.081)), a difference of 0.32 points. The random component of the model (participant, intercept) was significant (ICC = 0.890, $\chi^2(1) = 116$, $p < 0.0001$).

**Perceived difficulty and performance.** • Accuracy

**Table 9. Older adults, high task demands.**

| Region | Hemi. | B.A. | x | y | z | Z | k |
|---|---|---|---|---|---|---|---|
| Inf. occipital gyrus | R | 18 | −27 | −88 | −4 | 8.05 | 12010 |
| Mid. occipital gyrus | L | 19 | 36 | −88 | −1 | 7.68 | — |
| Sup. occipital gyrus | R | 19 | −27 | −64 | 29 | 7.35 | — |
| — | R | 7 | −24 | −61 | 41 | 7.01 | — |
| Inf. frontal oper. | R | 44 | −36 | 5 | 29 | 6.81 | — |
| Inf. parietal cortex | R | 40 | −33 | −46 | 44 | 6.63 | — |
| Supp. motor area | R | 6 | −6 | 8 | 53 | 6.56 | — |
| Precentral gyrus | L | — | 36 | −16 | 50 | 6.33 | — |
| Sup. parietal cortex | L | 7 | 24 | −61 | 47 | 6.18 | — |
| Inf. occipital gyrus | R | 19 | −39 | −67 | −13 | 6.17 | — |

Hemi: hemisphere; B.A.: Brodmann area; $k$: cluster size,—: a local maximum within a larger cluster. Results are corrected for multiple comparisons using threshold-free cluster enhancement, family-wise error correction and 1,000 resamples.

**Table 10.  Region of interest analyses.**

| Cluster number | Region of activation | Estimates | | | Estimated Marginal Means | | | |
|---|---|---|---|---|---|---|---|---|
| | | Age (Older— Younger) | Task demand (High—Low) | Age*Task demand | Younger, Low | Younger, High | Older, Low | Older, High |
| 1 | Left IFG (pars triangularis, orbitalis & opercularis), insula, OFC & precentral gyrus | −0.694 | 0.185 | 0.185 | 1.78* | 1.78* | 0.900* | 1.085* |
| | | −0.190 | −0.0333 | −0.146 | 0.134 | 0.247 | 0.0895 | 0.0562 |
| | | −0.105 | 0.140 | −0.145 | 0.161 | 0.445 | 0.200 | 0.3399 |
| | | −0.398 | −0.104 | −0.0486 | 0.843* | 0.788* | 0.493* | 0.3892 |
| | | 0.133 | 0.0555 | 0.0291 | −0.127 | −0.100 | −0.0228 | 0.0327 |
| | | −0.110 | 0.0304 | 0.0189 | 0.370 | 0.382 | 0.242 | 0.2719 |
| | | −0.151 | −0.00550 | 0.0319 | 0.107 | 0.0696 | −0.0755 | −0.081 |
| | | 0.848 | −0.0162 | −0.0165 | 1.42* | 1.42* | 2.28* | 2.27* |
| | | −0.310 | 0.0265 | −0.0128 | 0.420 | 0.459 | 0.122 | 0.148 |
| 2 | Left pMTG, pITG & pFG | −0.129 | 0.118 | 0.0182 | 0.775* | 0.875* | 0.628* | 0.746* |
| | | −0.120 | 0.0971 | 0.0278 | 1.09* | 1.16* | 0.942* | 1.04* |
| | | 0.321 | −0.0346 | 0.0391 | 1.33* | 1.26* | 1.61* | 1.58* |
| | | 0.133 | −0.00400 | 0.0252 | 0.103 | 0.0735 | 0.210 | 0.206 |
| | | 0.641 | −0.0658 | 0.106 | 0.205 | 0.0331 | 0.740* | 0.674* |
| 3 | Bilateral dmPFC | −0.362 | 0.0463 | 0.0481 | 1.58* | 1.57* | 1.17* | 1.21* |
| | | −0.0256 | −0.132 | 0.173 | 0.963* | 0.658* | 0.764* | 0.632* |
| | | −0.138 | −0.00830 | 0.0754 | 2.28* | 2.19* | 2.06* | 2.05* |
| 4 | Right IFG (pars orbitalis) & insula | −0.117 | −0.0417 | 0.0203 | 0.841* | 0.779* | 0.703* | 0.662* |
| | | 0.214 | −0.2074* | 0.0214 | −0.157 | −0.386* | 0.0352 | −0.172 |
| 5 | Right IFG (pars triangularis) | −0.169 | −0.3413* | 0.0133 | 0.931* | 0.577 | 0.749* | 0.408 |

Bold* = $p < 0.05$ uncorrected; IFG = inferior frontal gyrus; OFC = orbitofrontal cortex; p = posterior; ITG = inferior temporal gyrus; MTG = middle temporal gyrus; FG = fusiform gyrus; dmPFC = dorsomedial prefrontal cortex. Estimates are the parameters estimates calculated by the mixed-effects model and the calculated marginal means for the high versus low conditions of the semantic task.

In predicting accuracy using perceived difficulty, age group, and sex, there was no significant interaction between age group and perceived difficulty ($F(1, 147) = 0.0111$, $p = 0.916$). There were no main effects of age group ($F(1, 147) = 0.368$, $p = 0.545$), perceived difficulty ($F(1, 147) = 3.61$, $p = 0.0595$), or sex ($F(1, 147) = 0.143$, $p = 0.706$) on accuracy. The random component of the model (participant, intercept) was not significant (ICC = 0.00, $\chi^2(1) = 0.0$, $p = 1.0$).

• Response time

Finally, in predicting RTs using perceived difficulty, age group, and sex, there were significant main effects of perceived difficulty ($F(1, 103) = 24.7$, $p < 0.0001$) and age group ($F(1, 75.7) = 11.0$, $p = 0.00137$). The main effect of sex ($F(1, 72.8) = 0.426$, $p = 0.516$) and the interaction between age group and perceived difficulty ($F(1, 103) = 0.0580$, $p = 0.810$) were not significant. Greater perceived difficulty was associated with longer RTs ($\beta = 102$, one unit increase in perceived difficulty was associated with an increase in response times of 102 ms). Older age was associated with longer response times (older adults, mean (standard error) = 2243 (48.8); younger adults, mean (standard error) = 2121 (46.6)). The random component of the model (participant, intercept) was significant (ICC = 0.953, $\chi^2(1) = 177$, $p < 0.0001$).

**Perceived difficulty and brain activation in regions of interest.**   No regions were significantly activated as a result of perceived difficulty, as Table 11 shows.

**Table 11. Region of interest analyses using perceived difficulty.**

| Cluster number | Region of activation | Estimates | | |
|---|---|---|---|---|
| | | Age (Older—Younger) | Task demand (High—Low) | Age* Likert |
| 1 | Left IFG (pars triangularis, orbitalis & opercularis), insula, OFC & precentral gyrus | −1.358 | −0.163 | 0.263 |
| | | 0.590 | 0.013 | −0.253 |
| | | 0.485 | −0.135 | −0.125 |
| | | −0.058 | 0.153 | −0.146 |
| | | −0.143 | 0.030 | 0.115 |
| | | 0.049 | 0.060 | −0.062 |
| | | −0.409 | 0.025 | 0.089 |
| | | 1.037 | 0.111 | −0.095 |
| | | −0.092 | 0.047 | −0.084 |
| 2 | Left pMTG, pITG & pFG | 0.327 | 0.109 | −0.165 |
| | | −0.087 | −0.027 | −0.004 |
| | | 0.127 | 0.107 | 0.061 |
| | | 0.067 | 0.130 | 0.018 |
| | | 0.180 | 0.342 | 0.093 |
| 3 | Bilateral dmPFC | −0.562 | −0.093 | 0.102 |
| | | −0.445 | 0.289 | 0.080 |
| | | 0.156 | 0.092 | −0.137 |
| 4 | Right IFG (pars orbitalis) & insula | −0.298 | 0.136 | 0.041 |
| | | −0.316 | 0.007 | 0.197 |
| 5 | Right IFG (pars triangularis) | −1.235 | 0.213 | 0.327 |

IFG = inferior frontal gyrus; OFC = orbitofrontal cortex; p = posterior; ITG = inferior temporal gyrus; MTG = middle temporal gyrus; FG = fusiform gyrus; dmPFC = dorsomedial prefrontal cortex. Estimates are for the relationship between perceived difficulty and brain activity for the semantic task.

## Exploratory analyses

We had hypothesized that older adults who had high levels of brain activation in left-lateralized semantic control regions during the high-demand condition, similar to the younger adults, would have higher levels of task performance (reduced errors and RTs) than their counterparts whose brain activation was lower in these regions, in accordance with the CRUNCH model. This would indicate that they had not yet reached their crunch point, after which performance and activation would decline. To accept this hypothesis, at least one of the ROIs mentioned would have to be activated. In Table 12, we can see ROIs that were significantly activated as a function of RT, namely, the IFG and the dmPFC bilaterally. In Table 13, we can see that no ROIs were significantly activated as a function of accuracy.

## Discussion

In this study, we examined the CRUNCH hypothesis by administering a semantic processing task with two levels of task demand to 39 younger and 39 older adults. We used a novel task that varied task demands (low versus high). Our younger and older participants were matched in terms of level of education, performance on questionnaires assessing engagement in cognitively stimulating activities, and performance on the MoCA and WAIS-III. The behavioral results confirmed that the task was successful at manipulating cognitive demands. Overall, accuracy rates were not affected by age group, perceived difficulty or sex. Response times were

**Table 12. Region of interest analyses and response time.**

| Cluster number | Region of activation | Estimates | | | | | | |
|---|---|---|---|---|---|---|---|---|
| | | Task demand (High—Low) | Age (Older—Younger) | RT | Task demand *RT | Task demand *Age | Age *RT | Task demand *Age *RT |
| 1 | Left IFG (pars triangularis, orbitalis & opercularis), insula, OFC & precentral gyrus | −0.092 | −0.887 | −1.113 | 0.081 | −0.008 | 0.103 | 0.066 |
| | | −0.108 | −0.982 | −1.288 | 0.143 | −0.684 | 0.522 | 0.207 |
| | | −0.441 | 0.247 | −1.132 | 0.369 | 0.135 | 0.038 | −0.145 |
| | | 0.205 | 0.313 | −0.940 | −0.088 | −1.131 | −0.217 | 0.473 |
| | | −0.676 | −2.907 | **−1.811*** | 0.386 | 1.237 | 1.489 | −0.603 |
| | | −0.487 | −1.955 | **−1.795*** | 0.289 | 0.418 | 0.956 | −0.235 |
| | | 0.189 | −0.732 | −0.542 | −0.087 | 0.223 | 0.283 | −0.082 |
| | | −0.540 | −1.683 | **−2.119*** | 0.319 | −0.823 | 1.264 | 0.292 |
| | | −0.188 | −1.229 | −0.928 | 0.135 | −1.551 | 0.483 | 0.633 |
| 2 | Left pMTG, pITG & pFG | −0.737 | 0.364 | −0.543 | 0.407 | 1.424 | −0.146 | −0.628 |
| | | 0.239 | −1.415 | −0.552 | −0.061 | −0.328 | 0.607 | 0.138 |
| | | 0.022 | −0.569 | −0.700 | −0.020 | −0.581 | 0.447 | 0.248 |
| | | −1.046 | −1.104 | −0.947 | 0.500 | 0.788 | 0.631 | −0.380 |
| | | −0.527 | −2.139 | −1.428 | 0.211 | 0.509 | 1.291 | −0.225 |
| 3 | Bilateral dmPFC | −0.549 | −3.661 | −2.380 | 0.330 | −0.438 | 1.621 | 0.134 |
| | | −0.213 | −2.884 | **−2.538*** | 0.040 | −0.510 | 1.384 | 0.240 |
| | | 0.274 | −1.966 | **−1.827*** | −0.104 | −1.130 | 0.913 | 0.490 |
| 4 | Right IFG (pars orbitalis) & insula | 0.857 | 0.016 | **−1.085*** | −0.384 | −1.000 | 0.018 | 0.458 |
| | | 0.172 | 1.144 | −0.264 | −0.173 | −0.749 | −0.392 | 0.344 |
| 5 | Right IFG (pars triangularis) | −0.756 | −0.444 | −0.893 | 0.219 | 1.254 | 0.152 | −0.561 |

**Bold**\* = $p < 0.05$ uncorrected; IFG = inferior frontal gyrus; OFC = orbitofrontal cortex; p = posterior; ITG = inferior temporal gyrus; MTG = middle temporal gyrus; FG = fusiform gyrus; dmPFC = dorsomedial prefrontal cortex.

longer in older adults and when perceived difficulty was scored higher. Similarly, timeouts (delayed responses) increased in the high-demand condition.

There was no statistically significant difference in accuracy between younger and older participants regardless of the condition; in other words, no age effect and no perceived difficulty effect on accuracy. This is in line with previous publications that show that accuracy in semantic tasks is generally well preserved in older adults, given their more extensive experience with word use and larger vocabulary than younger ones [1, 5, 8, 10–13]. In terms of response times, there was a statistically significant difference between younger and older participants; older adults were slower to respond in general (a mean difference of 169 ms). Like age group, perceived difficulty also had a main effect on RTs. This is in accordance with studies showing that older adults' RTs tend to be longer than younger ones' RTs [10]. Thus, the semantic memory task was successful at (1) manipulating task difficulty across two levels of demand, as shown in differences in accuracy, RTs, timeouts and perceived difficulty between the two conditions; and (2) demonstrating age-invariant behavioral performance by the older group (i.e., maintained in comparison with the younger adults' performance, while task demands can still be met by older adults who have not yet reached their CRUNCH point), as required to test the CRUNCH model [21, 38]. We did not, however, find a significant interaction between task demand and age group, or between task demand and sex for either RTs or accuracy.

There seems to be a contradiction when we refer to well-maintained semantic memory and then aim to test a hypothesis that suggests that older adults will perform worse than younger ones in a semantic judgment task. It is generally thought that in healthy aging, performance in

**Table 13. Region of interest analyses and accuracy.**

| Cluster number | Region of activation | Estimates | | | | | | |
|---|---|---|---|---|---|---|---|---|
| | | Task demand (High—Low) | Age (Older—Younger) | RT | Task demand *RT | Task demand *Age | Age *RT | Task demand *Age *RT |
| 1 | Left IFG (pars triangularis, orbitalis & opercularis), insula, OFC & precentral gyrus | −3.056 | 1.333 | −2.984 | 3.545 | 0.228 | −2.397 | −0.377 |
| | | 1.319 | 7.148 | 2.206 | −1.294 | −2.954 | −8.073 | 2.667 |
| | | −1.780 | 6.326 | −0.959 | 2.476 | −2.870 | −6.935 | 2.651 |
| | | 0.134 | 1.928 | −0.123 | −0.229 | −1.200 | −2.543 | 1.144 |
| | | 0.465 | 2.814 | −0.760 | −0.569 | −1.219 | −2.944 | 1.149 |
| | | −0.788 | 2.124 | −1.460 | 0.893 | −0.013 | −2.448 | −0.314 |
| | | 0.122 | 1.517 | 0.317 | −0.167 | −0.820 | −1.909 | 0.864 |
| | | −2.203 | 1.045 | −0.927 | 2.668 | 3.993 | −0.205 | −5.014 |
| | | −0.665 | 3.585 | −1.321 | 0.773 | −2.398 | −4.331 | 2.469 |
| 2 | Left pMTG, pITG & pFG | −1.140 | −2.890 | −2.487 | 1.347 | 3.827 | 3.222 | −4.462 |
| | | −0.098 | 3.282 | 0.438 | 0.245 | −2.405 | −3.838 | 2.627 |
| | | −1.003 | 1.304 | −1.965 | 0.997 | −0.681 | −1.064 | 0.721 |
| | | −0.919 | 0.006 | −1.819 | 0.966 | 1.050 | 0.218 | −1.301 |
| | | −2.173 | −2.541 | −3.370 | 2.215 | 4.150 | 3.542 | −4.737 |
| 3 | Bilateral dmPFC | −0.772 | −1.015 | −1.410 | 0.867 | 1.492 | 0.760 | −1.773 |
| | | −0.717 | −0.350 | −1.674 | 0.402 | 0.394 | 0.253 | −0.329 |
| | | −0.211 | 0.899 | −0.203 | 0.164 | −0.056 | −1.215 | 0.001 |
| 4 | Right IFG (pars orbitalis) & insula | 0.726 | 2.226 | −0.026 | −0.951 | −3.352 | −2.637 | 3.890 |
| | | 0.422 | 3.650 | 0.911 | −0.718 | −1.872 | −3.882 | 1.933 |
| 5 | Right IFG (pars triangularis) | −2.224 | 0.788 | −3.776 | 2.022 | −3.627 | −1.109 | 4.311 |

IFG = inferior frontal gyrus; OFC = orbitofrontal cortex; p = posterior; ITG = inferior temporal gyrus; MTG = middle temporal gyrus; FG = fusiform gyrus; dmPFC = dorsomedial prefrontal cortex.

tasks that require attention and control decrease, whereas tasks that depend on lifelong learning (such as semantic memory) are typically well maintained [21]. Thus, when compared with other cognitive functions in aging (e.g. attention, memory), semantic memory is well preserved [22, 93, 143] comparing autobiographical, episodic and semantic memory in young vs. older adults). When comparing semantic memory of older with younger adults, the literature has yielded numerous results in regards to neural activation increases or decreases, and performance (accuracy and response times), depending on the task utilized, inter-individual variability and the specific age group. For example, older adults are found to be performing equally to younger adults in semantic priming tasks [12] or tasks assessing vocabulary size [144]. However, more "tip of the tongue" phenomena are reported for older adults [53], reduced performance in older adults in a naming task [8] and increased response times when tasks involve semantic selection [22]. The answer may lie within the system of semantic memory which is thought to comprise of 2 or 3 sub-systems, and with each one of them differentially affected by aging. More specifically, a) semantic representations are thought to be well-maintained in aging, b) retrieval processes are thought to be age-invariant whereas c) semantic control processes are thought to be negatively impacted by aging [23].

The CRUNCH model was conceived on a working memory task of a quantifiable nature (number of items to retain) [36]. Studies that have tested CRUNCH with more than 2 difficulty levels, have done so on working memory [21, 38], including recently an evaluation of the model with 4 demand levels on working memory [40]. Our focus is on semantic memory

which is of a different nature, more context-dependent and more difficult to quantify or manipulate for task demands (see, for example, [47] on the salience of semantic features and [45] on the variability of strength between semantic representations). There is no previous study in our knowledge which has evaluated the CRUNCH model in the context of semantic memory with more than two levels of task demands. Studies on semantic memory that have examined the impact of differing task demands, have done so with 2 levels [50, 52, 62, 61, 64, 66].

The CRUNCH hypothesis derived from a test on working memory and refers to a continuum of task demands whereby at lower demands older adults perform equally with younger ones whereas at higher task demands, they can no longer deploy additional resources. Beyond this "crunch" point, brain activation reaches a plateau and performance drops. With working memory, it is possible to manipulate task demands, however, this is more challenging for semantic processing.

In addition to the lack of significant interaction between task demand and age group for RTs or accuracy, we also did not find the hypothesized age group by task demand interaction effect on brain activation, the crucial test of the CRUNCH model. Only the age group effect produced significant activation in the bilateral occipital cortex and the cerebellum, whereas no significant main effect of task demand condition was observed. Despite the lack of task demand effects and only minimal age effects, the experiment did demonstrate robust task-related brain activity. Independent of age, the semantic similarity judgment task activated a large bilateral frontotemporoparietal network. More specifically, different clusters of activation were observed when all task conditions were contrasted with the baseline, such as in the bilateral inferior frontal, parietal, supplementary motor, temporal and occipital brain regions. Overall, the activated regions correspond with some regions reported to belong to the semantic network. The semantic network is proposed to be composed of seven brain regions, mainly in the left hemisphere: posterior inferior parietal cortex, lateral temporal cortex, ventral temporal cortex, dorsomedial prefrontal cortex (dmPFC), inferior frontal gyrus (IFG), ventromedial prefrontal cortex, and posterior cingulate gyrus [27, 44]. In addition, the anterior temporal lobes (ATLs) bilaterally are believed to act as a semantic hub [47], and semantic control processes are underpinned by the PFC, pMTG, and dAG/IPS [55]. Semantic decisions, in particular, are reported to activate a large constellation of cortical regions, including bilaterally the ATLs, PFC, posterior temporal cortex and angular gyrus [27, 46, 145]. Our task did not activate the semantic network to the expected extent. Similarly, we did not find the expected activation in the ATLs which is proposed to act as a semantic hub.

ROI analyses highlighted uncorrected significant effects of task demands within the left and right IFG, the left pMTG, the pITG and the prefrontal gyrus. In the pars triangularis and the pars orbitalis, lower activity was observed for the high-demand level versus the low-demand level. This demonstrates the task demand effect in these regions. We did not find any significant interactions between task demand and activation in the ROIs. We found only minimal task demand effects and strong task-related effects. Similarly, we did not find that any ROIs were significantly activated as a result of perceived difficulty. During our exploratory analysis, we did not find any ROIs that were significantly activated as a function of accuracy. We found only that the IFG and the dmPFC bilaterally were activated as a function of response time.

Activation in the IFG and the pMTG has frequently been reported to be associated with the more difficult conditions of semantic tasks [45, 146] in terms of both the number of competing representations and the amount of semantic information to be retrieved [45, 146, 147]. Left IFG activation is also proposed to be modulated when semantic representations are competing with each other as well as in relation to the amount of information that must be retrieved [45, 148]. Noonan et al. [46] suggested that activation of the IFG and pMTG together is crucial in

establishing, maintaining and applying task-related, goal-related and contextual representations in semantic processing. The coactivation of these two ROIs is also associated with high executive-semantic demands [144]. Applying TMS to the pMTG has been found to interfere with semantic decisions [149], while TMS in the IFG or pMTG interferes with semantic retrieval (such as retrieval of weak semantic relationships) [150]. More specifically, it has been suggested that the left posterior IFG contributes to high-demand semantic decisions, whereas the right posterior IFG contributes to picture-based decisions [144]. Although the BA45 portion of the left mid-IFG appears to be strongly activated in the most difficult conditions of all tasks and input modalities (words or pictures) [68], other parts of the IFG have demonstrated differential specializations and activation depending on modalities and tasks [144]. Activation in the IFG has been found to be age-invariant in a semantic judgment task with two levels of difficulty and four across-the-lifespan age groups [64]. Thus, numerous accounts in the literature have provided support for the coactivation of the IFG and the pMTG for the controlled retrieval and management of semantic memory [25, 49, 55, 151], which our findings however did not confirm. Regarding age-related IFG activation, a meta-analysis on age-related changes in the neural networks supporting semantic cognition demonstrated reduced activation in the left IFG in older adults performing semantic tasks, whereas recruitment was enhanced in the right IFG, especially when their performance was not maintained in comparison to their younger counterparts [22]. Similarly, in a semantic judgment task, older adults were found to rely more on the left IFG when semantic competition was high [66]. Our finding of reduction in activation in semantic control region (IFG0 could be in line with CRUNCH's descending part of the inverted U-shaped relation between semantic processing demands and activation, whereby after a certain difficulty threshold, available resources have reached their maximum capacity, leading to reduced activation and a decline in performance.

The requirement for semantic control during semantic judgment tasks is still under investigation. Some studies have suggested that semantic judgment tasks require less effort for retrieval and control than naming tasks, for example [65], given the lower demand for mental imagery [14, 152, 153]. However, it has also been proposed that the requirement to select between competing information post-retrieval necessitates the recruitment of the semantic control processes to a larger extent, manifested as increased activation in the multiple demand network and specifically the left ventral PFC [26, 46, 65] and the left IFG, which is thought to be typically activated in semantic judgment tasks [64]. Nevertheless, overactivation in the IFG may simply reflect the fact that participants maintain triads in their working memory for longer while comparing their semantic features [61]. Overall, it is thought that semantic judgment tasks tap both the semantic and multiple demand networks extensively, as they require the integration of both internal and external representations [46]. Despite preserved behavioral performance, underactivation in the control-related IPC has been observed in the older adults as a result of increased task demands; the authors explained this phenomenon by positing that semantic judgment tasks generally require less semantic control [65], which could be relevant in our study.

Within our data, the lack of a significant difference in activation between the two task demand conditions might be explained by the fact that our stimuli did not capture differences sufficient to yield a difference in neurofunctional activation. For example, we did not control our stimuli for the living versus non-living feature, which has been shown to influence processing [119], nor did we control for motor or visual features [118]. It is possible that the task was not sufficiently challenging for either the younger or older participants, and thus did not require the recruitment of additional neural resources. Alternatively, it is possible that the difference between low and high task demands was not big enough to provoke an increase in activation in either age group. It is also possible that the task was already too demanding for both

groups, so that no additional activation was possible, since participants had taxed their neural resources to the limit. Indeed, because the IFG is key to semantic processing as part of the semantic network, demonstrating robust activation across numerous semantic tasks, its spare capacity for additional recruitment may be limited, in either younger or older adults [23], which may have been the case in our study.

Jamadar [40] recently tested the CRUNCH model using a visuospatial working memory task with four levels of task demands. The results showed an effect opposite to the one predicted by CRUNCH. Jamadar challenged the CRUNCH model as it cannot easily be tested or falsified based on imaging methodologies, since regardless of whether activation increases or decreases, it can still be claimed to be compensation, whether successful or failed (e.g., assuming that behavioral performance would be worse without the additional activation). She concluded that concepts such as cognitive reserve, brain maintenance, compensation and dedifferentiation face issues related to definition, operationalization and falsification. More specifically, to test CRUNCH, it is necessary to manipulate task demands parametrically across three or four levels; however, not all cognitive fields are amenable to such manipulations. This might be true of semantic memory, since defining task demand levels for our conceptual store knowledge is not as straightforward as it can be with working memory, for example.

Specific to semantic memory preservation in aging, the findings of studies mentioned above may be partly in line with two compensatory hypotheses described in the Introduction: the executive hypothesis refers to the recruitment of domain-general executive processes, seen as overactivation in prefrontal, dorsal premotor, inferior frontal and inferior parietal brain regions, to compensate for age-related cognitive decline [6, 101]. A recent meta-analysis found that activation shifts from neurally specialized regions to more task-general areas with aging [22]. For example, in a study where participants had to decide whether two words shared a common feature, better-performing older adults showed more activation than both younger adults and poorer-performing older adults, notably in the bilateral premotor cortex as well as the inferior parietal and lateral occipital cortex, which are important regions for executive functions and object knowledge [56]. In a semantic judgment task using MEG in which participants decided whether a word was concrete or abstract, older participants activated the pMTG, inferior parietal lobule and ATL more than younger ones, whereas the younger participants activated the left IPC more than the older ones, leading the authors to conclude that the older participants overactivated their executive control regions to compensate and maintain their performance.

Alternatively, the semantic hypothesis, also known as the LAPA effect, refers to the recruitment of supplementary semantic processes and representations, demonstrated in overactivation in older people in the left posterior temporoparietal cortex [102, 103]. Given that executive functions decline more than language during aging, this hypothesis assumes that semantic are more likely than executive processes to undergo compensatory recruitment [104]. For example, in a semantic judgment study where participants had to decide whether or not a word denoted an animal, older participants had more bilateral parietal, temporal and left fusiform activation than younger people, who presented more dorsolateral activation, leading the authors to suggest that older participants were compensating with more semantic processes whereas younger participants were relying on frontal-based executive strategies [105]. It is important to note, however, that regions such as the LIFG and PFC subserve both executive and language functions, as described in the semantic network model [53], and this may make it difficult to distinguish between semantic and executive effects. Additional analyses should elucidate the role of the semantic control network in younger and older adults performing a semantic judgment task with two levels of task demand.

The study used taxonomic relationships, which are based on common features and membership in taxonomic category vs. thematic relationships, which are based on co-occurrence in space and time [119]. Taxonomic relationships are feature-based however, not all features are equally important in establishing a relationship [154]. This selection process of features that are to be prioritized vs. ignored, and thus resolving the conflict of competing features, is part of the semantic similarity judgment [155, 156]. Semantic similarity judgment tasks are thought to be ideal to explore deep semantic processing as they are thought to work independently of memory processes [157]. More specifically, they are thought to tap on automatic processes and explicit access to semantic knowledge [61], which are thought to be more resistant to aging [158].

Semantic processing and related neural activation can be influenced by various semantic features [159]. Processing concrete versus abstract words for example provokes hemispheric asymmetries [61]. The "hedonic valence" base (degree of positive or negative affective association) has been reported to have an impact on semantic processing [160]. Highly imageable words engage the right hemisphere more [161], as they have richer semantic representations and activate more semantic features [162]. Different activation patterns can be found for words representing natural versus man-made artifacts [163] and action versus non-action words [164] among other word features. Words differing by age of acquisition have been found to produce different behavioral results [165]. In regards to preference for thematic or taxonomic relations, cultural differences are found in preferring one type of semantic processing over another (e.g. preference for thematic vs. taxonomic relationship by Asians vs. Americans) to resolve competing semantic judgment [117]. Similarly, many other factors are shown to affect semantic processing, such as living vs. non-living [119], motor, visual features, visual similarity or complexity [118], level of abstractness [166], all of them interplaying with task demands and can thus influence processing of these representations [143].

For this study, we aimed to control for some factors which affect processing, namely frequency, imageability and length. Controlling for all possible semantic features which have been shown to affect semantic processing would have been extremely challenging, a feature which is inherent in the study of semantic processing in general.

## Acknowledgments

We wish to thank Carollyn Hurst, André Cyr and Daniel Papp for their help with MRI acquisitions, Julien Cohen-Adad for revising the MRI acquisition parameters, Catherine Dubé for her administrative support, and Perrine Ferré for sharing the statistical maps that were used for power analysis.

## Author Contributions

**Conceptualization:** Niobe Haitas, Maximiliano Wilson.

**Data curation:** Niobe Haitas, Jade Dubuc, Camille Massé-Leblanc, Vincent Chamberland.

**Formal analysis:** Niobe Haitas, Mahnoush Amiri, Tristan Glatard, Jason Steffener.

**Funding acquisition:** Yves Joanette.

**Investigation:** Niobe Haitas, Yves Joanette.

**Methodology:** Niobe Haitas, Mahnoush Amiri, Tristan Glatard, Maximiliano Wilson, Jason Steffener.

**Project administration:** Niobe Haitas.

**Resources:** Yves Joanette.

**Software:** Jason Steffener.

**Supervision:** Niobe Haitas, Maximiliano Wilson, Yves Joanette, Jason Steffener.

**Validation:** Yves Joanette, Jason Steffener.

**Writing – original draft:** Niobe Haitas.

**Writing – review & editing:** Niobe Haitas, Jason Steffener.

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
