## [Decision Letter · Decision Letter 0]

9 Nov 2023

PONE-D-23-19620Registered report: Age-preserved semantic memory and the CRUNCH effect manifested as differential semantic control networks: an fMRI studyPLOS ONE

Dear Dr. Haitas,

Thank you for submitting your manuscript to PLOS ONE. As you know, the key purpose of Stage 2 Registered Report review is to verify that the registered protocol was followed, that any deviations from the protocol are clearly reported and explained, and that the conclusions are supported by the data. I have now received reviews from two experts in the field (one of them had also reviewed your original protocol submission) and both of the reviewers feel that you generally followed the protocol. Reviewer 2 identified two key conclusions that do not seem to follow from the data and noted some (minor) protocol deviations and exploratory (not pre-registered) analyses that were not clearly identified as exploratory. Reviewer 1 also had some concerns that can be addressed through clarification/revision. Their recommendations are quite clear and constructive, so I believe they should be straight-forward to address in a revision.

We look forward to receiving your revised manuscript.

Kind regards,

Daniel Mirman

Academic Editor

PLOS ONE

Journal Requirements:

5. Please include a copy of Table 1 which you refer to in your text on page 45.

6. We note that Figures 7 and 8 in your submission contain copyrighted images. All PLOS content is published under the Creative Commons Attribution License (CC BY 4.0), which means that the manuscript, images, and Supporting Information files will be freely available online, and any third party is permitted to access, download, copy, distribute, and use these materials in any way, even commercially, with proper attribution. For more information, see our copyright guidelines: http://journals.plos.org/plosone/s/licenses-and-copyright.

a. You may seek permission from the original copyright holder of Figures 7 and 8 to publish the content specifically under the CC BY 4.0 license. 

Reviewers' comments:

Reviewer's Responses to Questions

**Comments to the Author**

1. Does the manuscript adhere to the experimental procedures and analyses described in the Registered Report Protocol?

If the manuscript reports any deviations from the planned experimental procedures and analyses, those must be reasonable and adequately justified.

Reviewer #1: Yes

Reviewer #2: Yes

2. If the manuscript reports exploratory analyses or experimental procedures not outlined in the original Registered Report Protocol, are these reasonable, justified and methodologically sound?

A Registered Report may include valid exploratory analyses not previously outlined in the Registered Report Protocol, as long as they are described as such.

Reviewer #1: Yes

Reviewer #2: Yes

3. Are the conclusions supported by the data and do they address the research question presented in the Registered Report Protocol?

The manuscript must describe a technically sound piece of scientific research with data that supports the conclusions. The conclusions must be drawn appropriately based on the research question(s) outlined in the Registered Report Protocol and on the data presented.

Reviewer #1: Yes

Reviewer #2: No

4. Have the authors made all data underlying the findings in their manuscript fully available?

Reviewer #1: Yes

Reviewer #2: Yes

5. Is the manuscript presented in an intelligible fashion and written in standard English?

Reviewer #1: Yes

Reviewer #2: Yes

6. Review Comments to the Author

Please use the space provided to explain your answers to the questions above. (Please upload your review as an attachment if it exceeds 20,000 characters)

Reviewer #1: I was a reviewer of this article at the proposal phase. The study aims at investigating age-related differences in semantic memory versus semantic control under the CRUCNH framework. Participants performed a semantic decision task where a target item had to be linked to one of two options. Task difficulty is varied by manipulating the semantic distance between the target word and the options. Despite the predictions of CRUNCH, the results did not show differential brain activation across groups as a function of task demands (i.e., increasing brain activity with increasing task demands for younger adults, but a decline (or plateauing) of brain activity with increasing task demands for older adults).

I think this paper is based on some solid experimental design and control, and that the results would be of great interest to the researchers in the field. I particularly liked the perceived vs. actual (manipulated) difficulty as it takes care of a whole host of confounding lexical variables that could have potentially affected the results (e.g., frequency, phonological similarity, orthographic similarity, etc.).

I have a couple of minor concerns that I encourage the authors to discuss in a revised version, and a couple of suggestions that the authors might want to consider either in the current manuscript or in future research.

Concerns:

1. It is difficult to know what the participants base their decisions on during the semantic judgement task. Do they consider features such as size, color, texture, animacy, concreteness etc.? Importantly, it is possible that the “high demand” condition is (perceived to be) more difficulty simply because there are more variables that can go into the judgement (e.g. semantic relatedness + size, color, texture ...). I think a discussion of this concern would be helpful for the reader.

2. On page 34, the authors say: “Based on the above pilot data, we were able to confirm that our task generated an effect of task demands that impacted task performance (accuracy and RTs) differently in younger and older adults, in the expected directions”. This is a little misleading because they did not find an Age:Task Demand interaction. They do see differential impacts of task on accuracy vs. RTs, with accuracy being equal across groups, but RTs being significantly longer for older adults. But this sentence is phrased in a way that implies the existence of an Age:Task Demand interaction, which is not true.

Suggestions:

1. Currently, the authors manipulate semantic distance using a norming study. Another (maybe more powerful way) would be to use tools such as GloVe (Global Vectors) for word representations (Pennington et al., 2014) to compute semantic similarity between the pairs in each triad. The vector representations in GloVe (and other similar NLP tools) are calculated based on really large corpora and may therefore offer more valid/stable distance values.

2. The semantic similarity is one way of varying task difficulty, but another way to look at the data is to use RTs themselves as a measure of task difficulty. Although RTs are the dependent variable, and the conditions probably correlate with RTs, there might still be variations in brain activity that is explained by RTs, but not by task demands. This is because there might still be quite a bit of variation within each demand condition. RTs can be viewed as an index for “actually experienced” difficulty, with longer RTs denoting more difficulty. One can then use this new variable (either categorical or continuous) as a predictor of brain activity. I wonder if the results would change when analyzed this way.

Typos:

Page 13: “should to”  “should”

Page 63: “said”  “set”?

Reviewer #2: Protocols in the Stage 1 protocol have been followed. However, I have some major concerns about how the results are summarised in the Discussion (see the last two comments below).

The method states that behavioural data were analysed with mixed-effects models and not repeated-measures ANOVA as stated in the Stage 1 protocol. However, the results reported seem to follow the Stage 1 approach. I suggest returning the Methods text to original pre-registered protocol.

I suggest reporting mean RTs and untransformed accuracies for each of the four conditions, given the importance of these data for the interpretation of study results.

fMRI analyses: no information about the thresholding approach was given in the Stage 1 protocol, which is unfortunate but cannot now be corrected. The thresholding approach used is reasonable but alpha should reported (i.e., is it p<0.05 with correction for multiple comparisons?).

What are the direction of the effects in Figure 8? Are the age effects voxels that were more active in old people or in young people? The condition effects – are these voxels that were more active for the semantic task or the control task? I ask because the activated areas don’t look quite like typical semantic regions.

What do the numbers in Table 8 represent? Are the estimated means for the effects of semantic vs. control task? A more informative caption is needed here, and in Table 9.

The timeout analyses were not pre-registered. They should be labelled as exploratory and some rationale given for them.

Pg. 62 “the semantic memory task was successful at… (2) demonstrating age-invariant behavioral performance by the older group (i.e., maintained in comparison with the younger adults’ performance), as required to test the CRUNCH model."

This is at odds with the author’s own predictions on page 23, where they predict that young people would be more accurate than young people (at least in the high demand condition). Some explanation is needed here – if parity in accuracy was needed to test the CRUNCH model, why did they design a study where they expected young people to be more accurate?

Pg. 62-63. Most of the regions described as semantic here are not activated in Figure 8 (i.e., posterior inferior parietal cortex, lateral temporal cortex, ventromedial prefrontal cortex and posterior cingulate gyrus). Nor are the ATLs. To suggest that these areas are activated in their study is highly misleading. At best they have some IFG and dmPFC, plus some posterior ventral temporal around the visual wordform area. A semantic network this is not.

Pg 63 “In the pars triangularis and the pars orbitalis, lower activity was observed for the high-demand level versus the low-demand level. This demonstrates the task demand effect in these regions.” This is very worrying because there should be *higher* activity in the high-demand condition here, not lower. Either this is a major typo or the experiment has delivered the opposite results to what we’d expect from the semantic control literature (as reviewed in detail in the next paragraph!). This needs to be addressed.

7. PLOS authors have the option to publish the peer review history of their article (what does this mean?). If published, this will include your full peer review and any attached files.

Reviewer #1: No

Reviewer #2: No

---

## [Author Response · Author response to Decision Letter 0]

23 Feb 2024

Response to reviewers

Reviewer #1

Concerns:

1. It is difficult to know what the participants base their decisions on during the semantic judgement task. Do they consider features such as size, color, texture, animacy, concreteness etc.? Importantly, it is possible that the “high demand” condition is (perceived to be) more difficulty simply because there are more variables that can go into the judgement (e.g. semantic relatedness + size, color, texture ...). I think a discussion of this concern would be helpful for the reader. 

We added some reflections in the discussion. 

The study used taxonomic relationships, which are based on common features and membership in taxonomic category vs. thematic relationships, which are based on co-occurrence in space and time (Sachs et al. 2008). Taxonomic relationships are feature-based however, not all features are equally important in establishing a relationship (Schmidt et al. 2012). This selection process of features that are to be prioritized vs. ignored, and thus resolving the conflict of competing features, is part of the semantic similarity judgment (Miller et al. 1991). Semantic similarity judgment tasks are thought to be ideal to explore deep semantic processing as they are thought to work independently of memory processes (Evans et al. 2012, Reilly and Peelle 2008). More specifically, they are thought to tap on automatic processes and explicit access to semantic knowledge (Sabsevitz 2005), which are thought to be more resistant to aging (Wlotko, Leet and Federmeier 2010). 

Semantic processing and related neural activation can be influenced by various semantic features (Pexman, Lupke, & Hino, 2002). Processing concrete versus abstract words for example provokes hemispheric asymmetries (Demonet et al., 2005; Sabsevitz et al., 2005), probably because concrete words activate both a verbal and a nonverbal code and have more semantic features as opposed to abstract words which are processed mainly verbally (Hagoort, 1998). The ‘hedonic valence’ base (degree of positive or negative affective association) has been reported to have an impact on semantic processing (Vigliocco et al., 2013). Highly imageable words engage the right hemisphere more (Nocentini et al., 2001), as they have richer semantic representations and activate more semantic features (G. a. L. Evans, Lambon Ralph, & Woollams, 2012; Murphy, 1990; Pexman et al., 2002; Sabsevitz et al., 2005). Different activation patterns can be found for words representing natural versus man-made artifacts (Dilkina & Lambon Ralph, 2012; Fuggetta et al., 2009; Hagoort, 1998) and action versus non-action words (Papeo, Pascual-Leone, & Caramazza, 2013) among other word features. Words differing by age of acquisition have been found to produce different behavioral results (Cortese & Khanna, 2007; Wilson, Cuetos, Davies, & Burani, 2013). In regards to preference for thematic or taxonomic relations, cultural differences are found in preferring one type of semantic processing over another (e.g. preference for thematic vs. taxonomic relationship by Asians vs. Americans) to resolve competing semantic judgment (Gutchess, Hedden, Ketay, Aron and Gabrieli 2010). Similarly, many other factors are shown to affect semantic processing, such as living vs. non-living, (Sachs 2008), motor, visual features, visual similarity or complexity (Geng 2016), level of abstractness (Grieder et al. 2012), all of them interplaying with task demands and can thus influence processing of these representations (Sabsevitz, Binder 2011, Meteyard 2012, Barsalou 2008). 

For this study, we aimed to control for some factors which affect processing, namely frequency, imageability and length. Controlling for all possible semantic features which have been shown to affect semantic processing would have been extremely challenging, a feature which is inherent in the study of semantic processing in general. 

2. On page 34, the authors say: “Based on the above pilot data, we were able to confirm that our task generated an effect of task demands that impacted task performance (accuracy and RTs) differently in younger and older adults, in the expected directions”. This is a little misleading because they did not find an Age: Task Demand interaction. They do see differential impacts of task on accuracy vs. RTs, with accuracy being equal across groups, but RTs being significantly longer for older adults. But this sentence is phrased in a way that implies the existence of an Age: Task Demand interaction, which is not true. 

We did not find an interaction indeed, as this sentence may imply. We corrected the sentence in the manuscript. 

Suggestions:

1. Currently, the authors manipulate semantic distance using a norming study. Another (maybe more powerful way) would be to use tools such as GloVe (Global Vectors) for word representations (Pennington et al., 2014) to compute semantic similarity between the pairs in each triad. The vector representations in GloVe (and other similar NLP tools) are calculated based on really large corpora and may therefore offer more valid/stable distance values. 

Thank you for this recommendation. It would have been great if we had been aware that such a tool was available at the time the stimuli was created. Unfortunately, we had difficulty to find such norming studies/corpora on French words at the time. We hope to use GloVe in future research.

2. The semantic similarity is one way of varying task difficulty, but another way to look at the data is to use RTs themselves as a measure of task difficulty. Although RTs are the dependent variable, and the conditions probably correlate with RTs, there might still be variations in brain activity that is explained by RTs, but not by task demands. This is because there might still be quite a bit of variation within each demand condition. RTs can be viewed as an index for “actually experienced” difficulty, with longer RTs denoting more difficulty. One can then use this new variable (either categorical or continuous) as a predictor of brain activity. I wonder if the results would change when analyzed this way. 

It would be an interesting methodology to use response times (RTs) as a measure of task difficulty and brain activation. This methodology has been used in the literature, for example Davey et al. (2015) conducted an analysis predicated on the hypothesis that tasks requiring more controlled retrieval would likely exhibit longer response times and lower accuracy compared to tasks involving more automatic semantic judgments. Consistent with these expectations, the study found that response efficiency in baseline sessions, calculated as RT divided by the proportion of correct trials, was inferior for weak thematic associations as opposed to strong ones. Similarly, Noppeney and Price (2004) differentiated their conditions with respect to response times, by modeling trials separately with reaction times that were 1.25 SD above or below the mean, and thus differentiated the conditions easy vs. difficult based on actual performance of participants. Analyzing brain activity in relation to RTs may reveal regions that are more activated during high-demand conditions, indicating increased cognitive effort for example. We hope to pursue such an analysis as part of future research.

Typos:

Page 13: “should to”  “should” 

Page 63: “said”  “set”? 

We corrected the typos in the manuscript.

References

Davey J, Cornelissen PL, Thompson HE, Sonkusare S, Hallam G, Smallwood J, Jefferies E. Automatic and Controlled Semantic Retrieval: TMS Reveals Distinct Contributions of Posterior Middle Temporal Gyrus and Angular Gyrus. J Neurosci. 2015 Nov 18;35(46):15230-9. doi: 10.1523/JNEUROSCI.4705-14.2015. PMID: 26586812; PMCID: PMC4649000

Miller, G. A., & Charles, W. G. (1991). Contextual correlates of semantic similarity. Language and Cognitive Processes, 6(1), 1-28.

Noppeney U, Price CJ. Retrieval of abstract semantics. Neuroimage. 2004;22: 164–70

Reviewer #2

Protocols in the Stage 1 protocol have been followed. However, I have some major concerns about how the results are summarised in the Discussion (see the last two comments below).

The method states that behavioural data were analysed with mixed-effects models and not repeated-measures ANOVA as stated in the Stage 1 protocol. However, the results reported seem to follow the Stage 1 approach. I suggest returning the Methods text to original pre-registered protocol. 

We corrected the method in the manuscript.

I suggest reporting mean RTs and untransformed accuracies for each of the four conditions, given the importance of these data for the interpretation of study results. 

We reported these in the manuscript.

fMRI analyses: no information about the thresholding approach was given in the Stage 1 protocol, which is unfortunate but cannot now be corrected. The thresholding approach used is reasonable but alpha should reported (i.e., is it p<0.05 with correction for multiple comparisons?). 

We reported these in the manuscript.

What are the direction of the effects in Figure 8? Are the age effects voxels that were more active in old people or in young people? The condition effects – are these voxels that were more active for the semantic task or the control task? I ask because the activated areas don’t look quite like typical semantic regions. 

The age effect represents increased brain activity for old people compared to young groups. The low results are for low task demands greater than the control condition. The high results are for high task demands greater than the control condition.

What do the numbers in Table 8 represent? Are the estimated means for the effects of semantic vs. control task? A more informative caption is needed here, and in Table 9. 

Estimates are the parameters estimates calculated by the mixed-effects model and the calculated marginal means for the high versus low conditions of the semantic task. For table 9 (now 10), estimates are for the relationship between perceived difficulty and brain activity for the semantic task. We reported these in the manuscript.

The timeout analyses were not pre-registered. They should be labelled as exploratory and some rationale given for them. 

The number of timeouts, trials with no response, increased with task demands. This unanticipated observation was not included in the registered report making the following results exploratory. We reported this in the manuscript.

Pg. 62 “the semantic memory task was successful at… (2) demonstrating age-invariant behavioral performance by the older group (i.e., maintained in comparison with the younger adults’ performance), as required to test the CRUNCH model." This is at odds with the author’s own predictions on page 23, where they predict that young people would be more accurate than young people (at least in the high demand condition). Some explanation is needed here – if parity in accuracy was needed to test the CRUNCH model, why did they design a study where they expected young people to be more accurate? 

We would like to point the reviewer to the authors’ response shared following feedback by reviewers at the pre-registration stage. 

The CRUNCH hypothesis refers to a continuum of task demands whereby at lower demands older adults perform equally with younger ones whereas at higher task demands, they can no longer deploy additional resources. Beyond this ‘crunch’ point, brain activation reaches a plateau and performance drops. The CRUNCH model was originally conceived on a working memory task of a quantifiable nature (number of items to retain) (Cappell et al., 2010). Studies that have tested CRUNCH with more than 2 difficulty levels, have done so on working memory ((Fabiani, 2012; Rypma, Eldreth, & Rebbechi, 2007; Schneider-Garces et al., 2010), including recently an evaluation of the model with 4 demand levels on working memory (Jamadar, 2020). 

Our focus is to study the CRUNCH hypothesis on semantic memory which is of a different nature, more context-dependent and more difficult to quantify or manipulate for task demands (see for example Patterson et al. 2007 on the salience of semantic features and Badre et al. 2002 on the variability of strength between semantic representations). Studies on semantic memory that have examined the impact of differing task demands, have done so with 2 levels (Chiou et al., 2018; Davey et al., 2015; Kennedy et al., 2015; Noppeney & Price, 2004; Sabsevitz et al., 2005; Zhuang et al., 2016). There is no previous study in our knowledge which has evaluated the CRUNCH model in the context of semantic memory with more than two levels of task demands (see however the behavioral study of Fu et al. 2017). We have shared explanation in the discussion. 

Pg. 62-63. Most of the regions described as semantic here are not activated in Figure 8 (i.e., posterior inferior parietal cortex, lateral temporal cortex, ventromedial prefrontal cortex and posterior cingulate gyrus). Nor are the ATLs. To suggest that these areas are activated in their study is highly misleading. At best they have some IFG and dmPFC, plus some posterior ventral temporal around the visual wordform area. A semantic network this is not. 

We corrected this in the manuscript.

Pg 63 “In the pars triangularis and the pars orbitalis, lower activity was observed for the high-demand level versus the low-demand level. This demonstrates the task demand effect in these regions.” This is very worrying because there should be *higher* activity in the high-demand condition here, not lower. Either this is a major typo or the experiment has delivered the opposite results to what we’d expect from the semantic control literature (as reviewed in detail in the next paragraph!). This needs to be addressed. 

In pages 13-15 of the manuscript, findings in regards to semantic control network activation as a function of task demands are discussed. Overall, the impact of aging on semantic processing varies across different regions of the prefrontal cortex, including the IFG. ‘According to CRUNCH, a reduction in activation in semantic control region could be interpreted as falling within the descending part of the inverted U-shaped relation between semantic processing demands and fMRI activation [29]: after a certain difficulty threshold, available neural resources from the semantic or multiple-demand control network have reached their maximum capacity, leading to reduced activation and a decline in performance [30]. In line with CRUNCH, maintained performance could depend on the additional recruitment of semantic control network resources but only between certain difficulty thresholds, before which increasing activation is unnecessary or beneficial and after which performance declines’. We have added clarifications in the manuscript.

---

## [Decision Letter · Decision Letter 1]

27 Mar 2024

Registered report: Age-preserved semantic memory and the CRUNCH effect manifested as differential semantic control networks: an fMRI study

PONE-D-23-19620R1

Dear Dr. Haitas,

We’re pleased to inform you that your manuscript has been judged scientifically suitable for publication and will be formally accepted for publication once it meets all outstanding technical requirements.

Kind regards,

Daniel Mirman

Academic Editor

PLOS ONE

Additional Journal Comments:

**Please ensure that your final submission explicitly references your previously published protocol ("Age-preserved semantic memory and the CRUNCH effect manifested as differential semantic control networks: An fMRI study"), including the doi, in the references section.**

Reviewers' comments:

Reviewer's Responses to Questions

**Comments to the Author**

1. Does the manuscript adhere to the experimental procedures and analyses described in the Registered Report Protocol?

If the manuscript reports any deviations from the planned experimental procedures and analyses, those must be reasonable and adequately justified.

Reviewer #1: Yes

2. If the manuscript reports exploratory analyses or experimental procedures not outlined in the original Registered Report Protocol, are these reasonable, justified and methodologically sound?

A Registered Report may include valid exploratory analyses not previously outlined in the Registered Report Protocol, as long as they are described as such.

Reviewer #1: Yes

3. Are the conclusions supported by the data and do they address the research question presented in the Registered Report Protocol?

The manuscript must describe a technically sound piece of scientific research with data that supports the conclusions. The conclusions must be drawn appropriately based on the research question(s) outlined in the Registered Report Protocol and on the data presented.

Reviewer #1: Yes

4. Have the authors made all data underlying the findings in their manuscript fully available?

Reviewer #1: Yes

5. Is the manuscript presented in an intelligible fashion and written in standard English?

Reviewer #1: Yes

6. Review Comments to the Author

Please use the space provided to explain your answers to the questions above. (Please upload your review as an attachment if it exceeds 20,000 characters)

Reviewer #1: I read the responses to my remaining concerns, and I'm convinced by them. I think this paper makes meaningful contributions to the field.

7. PLOS authors have the option to publish the peer review history of their article (what does this mean?). If published, this will include your full peer review and any attached files.

Reviewer #1: No

---

## [Editor Report · Acceptance letter]

23 May 2024

PONE-D-23-19620R1 

PLOS ONE

Dear Dr. Haitas, 

I'm pleased to inform you that your manuscript has been deemed suitable for publication in PLOS ONE. Congratulations! Your manuscript is now being handed over to our production team.

Kind regards, 

on behalf of

Dr. Daniel Mirman 

Academic Editor

PLOS ONE